# TimeSuite: Improving MLLMs for Long Video Understanding via Grounded Tuning

Xiangyu Zeng[1,2]   Kunchang Li[3,2]   Chenting Wang[6,2]   Xinhao Li[1,2]   Tianxiang Jiang[5,2]
Ziang Yan[4,2]   Songze Li[7,2]   Yansong Shi[5,2]   Zhengrong Yue[6,2]   Yi Wang[2,8]
Yali Wang[3,2]   Yu Qiao[2]   Limin Wang[1,2,†]

[1]Nanjing University   [2]Shanghai AI Laboratory   [3]SIAT, Chinese Academy of Sciences   [4]Zhejiang University
[5]University of Science and Technology of China   [6]Shanghai Jiao Tong University   [7]Fudan University
[8]Shanghai Innovation Institute

XiangyuZeng2001@outlook.com   lmwang@nju.edu.cn

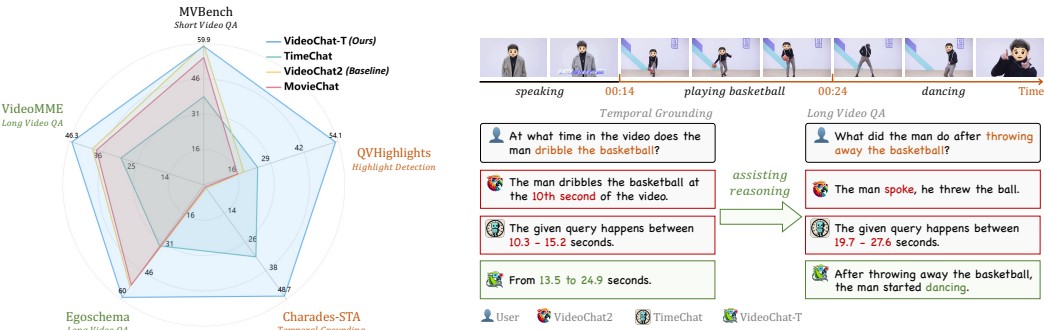

Figure 1: VideoChat-T demonstrates high performance for both long-form video question answering and temporal grounding. Our TimeSuite presents a collection of new designs to enhance the long video understanding capability of MLLMs. It will implicitly endow the MLLM with ability of correctly attending the visual segments when generating answers, thus relieving the hallucinations.

## ABSTRACT

Multimodal Large Language Models (MLLMs) have demonstrated impressive performance in short video understanding. However, understanding long-form videos still remains challenging for MLLMs. This paper proposes **TimeSuite**, a collection of new designs to adapt the existing short-form video MLLMs for long video understanding, including a simple yet efficient framework to process long video sequence, a high-quality video dataset for grounded tuning of MLLMs, and a carefully-designed instruction tuning task to explicitly incorporate the grounding supervision in the traditional QA format. Specifically, based on VideoChat, we propose our long-video MLLM, coined as VideoChat-T, by implementing a token shuffling to compress long video tokens and introducing Temporal Adaptive Position Encoding (TAPE) to enhance the temporal awareness of visual representation. Meanwhile, we introduce the TimePro, a comprehensive grounding-centric instruction tuning dataset composed of 9 tasks and 349k high-quality grounded annotations. Notably, we design a new instruction tuning task type, called Temporal Grounded Caption, to perform detailed video descriptions with the corresponding timestamps prediction. This explicit temporal location prediction will guide MLLM to correctly attend on the visual content when generating description, and thus reduce the hallucination risk caused by the LLMs. Experimental results demonstrate that our TimeSuite provides a successful solution to enhance the long video understanding capability of short-form MLLM, achieving improvement of **5.6%** and **6.8%** on the benchmarks of Egoschema and VideoMME, respectively. In addition, VideoChat-T exhibits robust zero-shot temporal grounding capabilities, significantly outperforming the existing state-of-the-art MLLMs. After fine-tuning, it performs on par with the traditional supervised expert models. Our code and dataset are available at https://github.com/OpenGVLab/TimeSuite.

---

[†] denotes the corresponding author.

# 1 INTRODUCTION

Multimodal Large Language Models (MLLMs) have demonstrated impressive video understanding performance by following the general human instructions to interpret the visual content (Li et al., 2023b; Zhang et al., 2023; Lin et al., 2023a; Jin et al., 2024; Wang et al., 2024e). However, these MLLMs still struggle in long video understanding, as a long video sequence may contain various dynamic actions and complex temporal relationships, making it difficult for MLLMs to effectively locate the key segments related to questions. When humans watch long videos, their attention is consciously focused on prominent segments, which may occur within a few seconds. NExT-GQA (Xiao et al., 2024) has also verified the relevance of temporal grounding for accurately answering video QA tasks. Therefore, a natural question arises: ***Can we enhance long video understanding by using temporal grounding as a auxiliary task?***

Previously, some works have made progress in temporal grounding task by using general MLLMs. They often enhance the temporal grounding capability of video MLLMs by designing specialized modules and perform specific supervised fine-tuning (Ren et al., 2024; Huang et al., 2024a;b). However, these overly specialized designs significantly impair the general QA capabilities of video MLLMs, resulting in great performance drop on the video QA task (as illustrated by TimeChat in Figure 1). Meanwhile, current research on long video understanding primarily focuses on architecture design, such as long-context LLMs (Liu et al., 2024a) and token compression (Song et al., 2024a). They can only capture holistic semantics in videos without the ability of localizing fine-grained information, leading to poor performance in temporal grounding tasks (as illustrated by MovieChat in Figure 1). So far, it is still challenging to build a video MLLM that is good at both tasks of temporal grounding and long video QA. We argue long video understanding could be assisted by explicitly performing temporal grounding, as grounding supervision enables MLLM to establish the detailed correspondance between the visual segments and fine-grained semantics. This fine-grained alignment would guide the MLLM to attend correctly video segments when generating answers and thus relieve the hallucination risk caused by the LLM.

Based on the above analysis, in this paper, we propose TimeSuite, a collection of new designs to improve the long video understanding capability of the existing short-form MLLMs, with a focus on incorporating grounding supervision in instruction tuning process. First, to address the high computational cost caused by the excessive number of visual tokens in long videos, we propose a simple Token Shuffle scheme to compress visual tokens, allowing the LLM to process more frame inputs. We also propose TAPE to generate adaptive position encodings, enhancing the temporal awareness of visual representations. The proposed structure does not introduce overly complex proprietary designs, which could be efficiently initialized with the parameters of short video MLLMs, without damaging the original performance of pre-trained MLLM. Second, to naturally incorporate the grounding ability into our MLLMs and yet still to preserve its original general QA capability, we design a new instruction tuning task, called Temporal Grounded Caption. This new task requires generating detailed segment-level description with corresponding timestamp prediction. Tuning on this new task will not only endow the MLLM with the extra grounding ability but also enhance its original long video QA performance, thanks to the requirement of building correspondence between grounded segments and detailed captions. Finally, we collect a comprehensive grounding-centric instruction tuning dataset for post-training our designed MLLMs, which is composed of 349K high-quality annotations covering 9 tasks. Based on this new dataset, we are able to perform grounded tuning with detailed captions on our proposed MLLMs (coined as VideoChat-T).

We verify the effectiveness of TimeSuite design through extensive experiments on the tasks of long video understanding and temporal grounding. VideoChat-T demonstrates a significant improvement in accuracy over baseline for long video understanding, with a 5.6% increase on Egoschema (Mangalam et al., 2023) and a 6.8% increase on VideoMME (Fu et al., 2024). Additionally, VideoChat-T exhibits robust zero-shot temporal localization capabilities on Charades-STA (Gao et al., 2017) and QVHighlights (Lei et al., 2021a). Our VideoChat-T outperforms the state-of-the-art temporal grounding MLLM of TimeChat from 50% to 100% for different metrics. After fine-tuning on the training set of temporal grounding benchmarks, the performance of VideoChat-T is on par with the state-of-the-art supervised expert models. The experiments demonstrate that *our VideoChat-T is the first end-to-end MLLM that is able to perform well on both temporal grounding and general video QA*. In particular, we show that grounded tuning with explicit location prediction can facilitate the long video understanding and relieve the hallucination risk.

## 2 RELATED WORK

**Video MLLMs.** With the advancement of open-sourced LLMs (Chiang et al., 2023; Touvron et al., 2023; Jiang et al., 2023), video MLLMs have emerged by utilizing projection bridges to link vision foundation models with LLMs (Li et al., 2023b; 2024b; Zhang et al., 2023; Li et al., 2024a). Limited by the training context length, thought these methods perform well with a small number of frame inputs, they meet significant challenges when processing long videos. The longer video length usually implies longer temporal relationships and more redundancies, resulting in the difficulty of extracting key clues (Zhou et al., 2024). Recently, several methods for long video handling have been proposed, such as exploiting long context LLM (Liu et al., 2024a; Zhang et al., 2024b; Xue et al., 2024; Wang et al., 2024d) and token compression (Li et al., 2023d; Song et al., 2024a; Zhang et al., 2024a) for enabling more visual inputs and agents for task decomposition or retrieval (Fan et al., 2024; Wang et al., 2024c;h). MovieChat (Song et al., 2024a) supports more frames by applying short-term and long-term memory to merge similar visual tokens. Yet, studies in learning objectives for long videos are less explored, making it difficult to alleviate the frequent hallucination of LLMs in long context reasoning. Our proposed TimeSuite leverages temporally-centric tasks to unlock the temporal perception potential of MLLMs, anchoring responses to the most relevant video segments.

**Temporal Grounding.** Temporal grounding is a fundamental capability in video understanding, associating semantics to specific clips with corresponding timestamps. Typical expert models (Lei et al., 2021b; Moon et al., 2023a;b; Lin et al., 2023b; Zeng et al., 2024) have been developed by formulating it into a timestamp regression from visual inputs and user queries. Most existing video MLLMs fail to address it compared with expert models, while some remedy its temporal grounding by specifically designed architectures and data (Huang et al., 2024a; Wang et al., 2024f; Li et al., 2024c; Wang et al., 2024g; Huang et al., 2024b; Qu et al., 2024). Timechat (Ren et al., 2024) binds visual features of images with timestamps and uses a sliding window to handle variable token length. From the perspective of training data, an instruction-tuning dataset TimeIT is constructed. Despite impressive improvements in temporal performance, these MLLMs still lag behind expert models and compromise general video dialogue capabilities. In this paper, we explore how to enhance the temporal grounding of MLLMs while preserving their original capabilities.

## 3 METHOD

In this section, we detail the proposed TimeSuite, a new collection of designs for improving short video MLLMs. Specifically, our TimeSuite includes a long video modeling framework, a high-quality video dataset for grounded tuning, and a carefully-designed instruction tuning task. With this new TimeSuite design, we are able to adapt the short-form video MLLM, obtaining significant performance improvements on two types of long video understanding tasks: traditional long video QA and temporal video grounding.

### 3.1 VIDEOCHAT-T

We first describe the architecture of our proposed long video modeling framework. Specifically, built upon VideoChat2 (Li et al., 2024b), we devise long-video version of VideoChat-T. Our VideoChat-T is composed of a video backbone for extracting visual representations, a visual-language connector to compress visual tokens and bridge the visual and languages modalities, a LLM to follow human instructions to interpret the video content.

The architecture of VideoChat-T is illustrated in Figure 2. Its workflow has three stages. In the first stage, long videos are evenly segmented into clips and the clips are embedded by the Video Encoder and Q-Former (Li et al., 2023a). Then, for compressing visual token number and highlighting crucial ones, token shuffling is employed to merge adjacent tokens, and TAPE is used to add temporal adaptive positional encodings. Finally, the compressed video token sequence is fed to the LLM to generate accurate responses that adhere to user requirements.

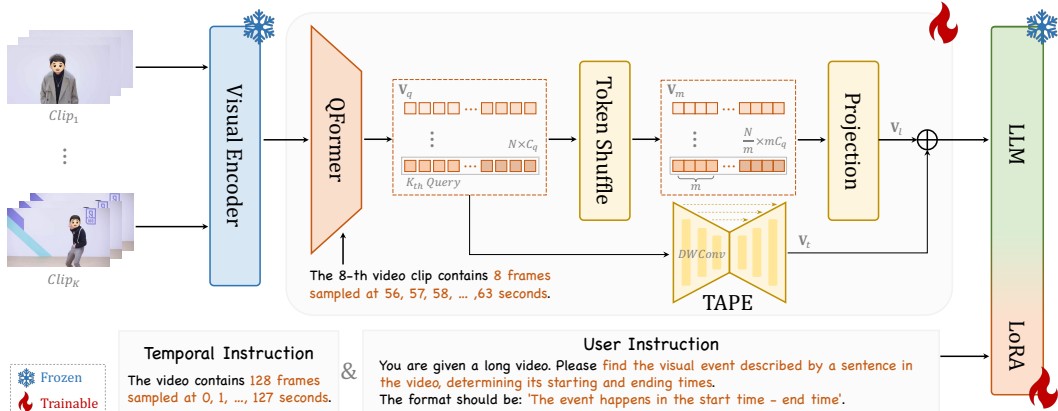

Figure 2: **Overall Architecture of VideoChat-T.** First, long videos are segmented into clips, which are then transformed into feature embeddings by video encoder and time-aware Qformer. Next, all visual tokens undergo Token Shuffle to compress overly long tokens, and generate adaptive positional encodings through TAPE. Finally, the long video tokens are concatenated with the user query, serving as the input of LLM, thereby generating appropriate responses.

### 3.1.1 BACKBONE DESIGN

**Video clip encoding.** For the given long video, we perform uniform sampling (Wang et al., 2019) to obtain $K \times T$ frames. We divide these frames into $K$ video segments in chronological order, and sample $T$ frames from each segment. Next, we use the video encoder and its visual-linguistic connector (Q-Former here) to encode each segment into $N$ tokens. After the aforementioned processing, the entire video is encoded into a sequence of visual tokens, denoted by $\mathbf{V}_q \in \mathbb{R}^{L \times C_q}$, where $C_q$ is the dimension of output token by the Q-Former and $L = K \times N$ is the total number of tokens for the entire video.

**Large Language Model.** According to previous research, images and visual cues are projected into the same feature space of the LLM. The LLM acts as an interaction interface in the MLLMs, being used to process multimodal inputs, parse user instructions, and generate appropriate responses. To afford the processing of long video sequence, we need to design an efficient compression module between the visual encoder and LLMs.

### 3.1.2 VL-CONNECTOR: TOKEN SHUFFLE

The increased number of sampled frames in long videos leads to a larger number of encoded visual tokens, causing a significant rise in the computational complexity and memory consumption of LLMs. Therefore, it is crucial to keep the number of visual tokens within an acceptable range. Some works have proposed various token compression schemes, such as clustering (Jin et al., 2024) and pooling (Huang et al., 2024b). However, clustering methods often struggle to maintain the temporal consistency, and pooling methods usually result in a certain loss of overall performance.

To address this, we propose a simple token shuffling compression scheme that ensures the temporal consistency of video tokens before and after compression while avoiding excessive performance loss. Previous methods often used a projector to achieve dimensional conversion. However, projecting visual encoding vectors from low to high dimensions does not increase information density. Therefore, we propose to rearrange multiple visual tokens along the channel dimension. Specifically, for the long video $\mathbf{V}_q = [v_q^1, v_q^2, ..., v_q^L] \in \mathbb{R}^{L \times C_q}$, we concatenate $m$ adjacent tokens along the channel dimension to obtain the reshaped visual feature $\mathbf{V}_m = [v_m^1, v_m^2, ..., v_m^{\frac{L}{m}}] \in \mathbb{R}^{\frac{L}{m} \times mC_q}$, where each merged token $v_m^i$ is represented as:

$$v_m^i = \text{Concat}(v_q^{(i-1)*m+1}, v_q^{(i-1)*m+2}, ..., v_q^{i*m}) \quad \forall i = 1, 2, ..., \tfrac{L}{m}.$$

Next, a linear projection layer is applied to the merged visual feature $\mathbf{V}_m$, generating the visual token sequences $\mathbf{V}_l \in \mathbb{R}^{\frac{L}{m} \times C_l}$ as input into the LLM, where $C_l$ represents the token channel di-

mension of the LLM. This scheme effectively reuses the projector of base model by replicating the original linear layer parameters $m$ times along the channel dimension, achieving an initialization equivalent to mean pooling with a window length of $m$. This design avoids introducing additional randomly initialized parameters that might disturb the original model, thus preserving the its original capabilities. Additionally, compared to directly using pooling, this method offers higher flexibility for fine-tuning to achieve better results (see ablation study, Table 4).

### 3.1.3 TEMPORAL ADAPTIVE POSITION ENCODING

To bind temporal positional information to visual tokens, we propose an adapter called **T**emporal **A**daptive **P**osition **E**ncoding (TAPE). Inspired by CPVT (Chu et al., 2021), our TAPE uses zero padding at both ends of the convolution as anchors, and gradually transmits relative positional encoding information. Without the need to add any special time tokens, TAPE can automatically perceive the relative temporal positions of the token sequence and generate temporal embeddings.

Specifically, the long video token sequence $\mathbf{V}_q$ is first compressed in the channel dimension by a linear layer and further compressed in sequence length by a pooling layer. Next, we use a U-Net-like structure composed of one-dimensional depthwise separable convolutions to progressively downsample the sequence, obtaining three one-dimensional temporal feature sequences with different resolutions. Subsequently, a convolution with a sufficiently long window is applied to the shortest temporal feature sequence, using zero padding at both ends as anchors to encode the relative temporal position of each token in the sequence (Chu et al., 2021). Then, we progressively upsample and restore the temporal feature sequences from short to long, using residual connections to retain temporal features at different scales. Finally, the temporal feature sequences are restored to the same length as $\mathbf{V}_l$ and aligned in the channel dimension by a linear layer, thereby obtaining the temporal features $\mathbf{V}_t$ output by the TAPE. For detailed implementation of TAPE, please refer to Appendix A.

Our proposed TAPE offers a plug-and-play module, which could be easily integrated into the network structure via residual connections, adding temporal position information to video tokens without disrupting the distribution of other trainable parameters. With appropriate training strategies, TAPE effectively preserves the model's generalization capabilities and enhances its temporal sensitivity (see ablation study, Table 3), which is important for temporal grounding task.

### 3.2 TIMEPRO: TEMPORAL GROUNDED INSTRUCTION DATA

Traditional temporal grounding datasets only contain monotonous ground truth, i.e., the start and end times of the target period. This data format performs well in training the classic expert models, but is difficult to unleash the potential of LLMs. Although several temporal grounding-centric datasets have been released for MLLM fine-tuning (Ren et al., 2024; Huang et al., 2024b), they still have deficiencies in data quantity, data quality, and task diversity. Thus, it is necessary to build a more comprehensive temporal dataset designed for the tuning of MLLMs.

Based on the criteria of diversity, length, and difficulty, we collect and clean several existing high-quality grounding-centric datasets (Ren et al., 2024; Huang et al., 2024a;b), and create two new datasets, resulting in the TimePro. Compared to previous temporal grounding-centric datasets, TimePro offers a larger volume of data, a broader distribution, and a higher task diversity, facilitating the learning of more generalizable temporal representations for MLLMs.

As shown in Figure 3(a), TimePro contains 9 task types from 15 datasets that are highly relevant to temporal grounding, containing approximately 349K high-quality temporal grounding annotations. The 9 tasks are specified as follows. **Temporal Video Grounding** involves identifying the start and end times of video content based on a natural language query (Anne Hendricks et al., 2017; Oncescu et al., 2021; Zala et al., 2023). **Dense Video Captioning** requires detecting events within a video and providing corresponding timestamps and descriptions (Krishna et al., 2017; Huang et al., 2020; Zhou et al., 2018). **Video Summarization** focuses on determining key frames or clips in the form of timestamps rather than semantic summaries (Song et al., 2015; Gygli et al., 2014). **Step Localization** aims to segment and describe important steps in a long video (Tang et al., 2019; Zala et al., 2023). **Transcribed Speech Generation** predicts speech content and its timestamps from visual signals (Zellers et al., 2022). **Reasoning Temporal Localization** combines timestamps with explanatory answers (Huang et al., 2024b). **Multi-format Temporal Grounding** includes single-turn and multi-turn dialogues with diverse question types (Huang et al., 2024a). **Highlight**

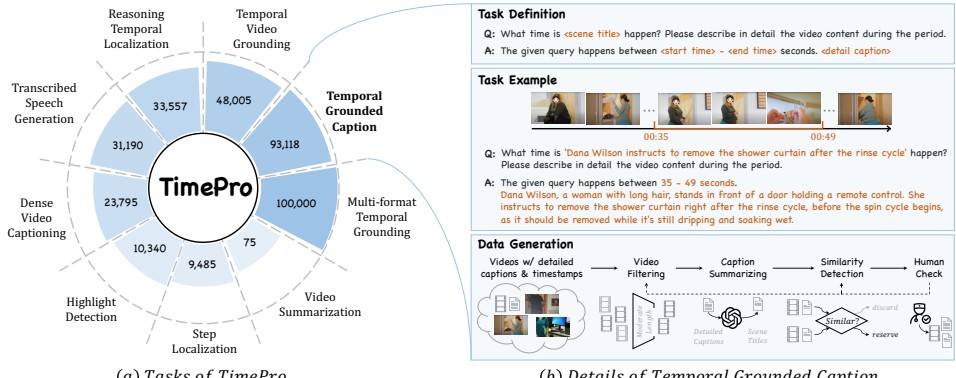

(a) Tasks of TimePro  (b) Details of Temporal Grounded Caption

Figure 3: (a) The proposed temporal centric instruction-tuning dataset, **TimePro**. This dataset contains approximately 349K high-quality and strongly temporally correlated data. (b) The proposed Temporal Grounded Caption fine-tuning data paradigm. It effectively reducing the occurrence of hallucinations. We employ a 4-stage processing pipeline to ensure the quality of the generated data.

**Detection** identifies the most significant moments in a video based on a query (Lei et al., 2021a). **Temporal Grounded Caption** uses a brief scene title to output both the time period and fine-grained description for the scene. More detailed information about TimePro is available in the appendix B. It should be noted that Temporal Grounded Caption is our newly-designed task that can help our model to establish fine-grained correspondence between visual segment and linguistic description.

## 3.3 TEMPORAL GROUNDED CAPTION TASK

Some studies have shown that MLLMs often exhibit severe hallucinations when dealing with fine-grained perception tasks (Ji et al., 2023; Huang et al., 2023; Golkar et al., 2023). Since our VideoChat-T directly regresses the timestamps corresponding to the text queries using MLLMs, it is more susceptible to hallucinations compared to methods that use external expert models as decoders (Wu et al., 2024). By forcing the video MLLMs to predict the event occurrence time and simultaneously describe the visual content evidence, we attempt to anchor these queries to the relevant time segments within the video, rather than generating hallucinations originating from LLM itself. Based on this analysis, we design the Temporal Grounded Caption task.

The top of Figure 3(b) illustrates the definition of Temporal Grounded Caption. We use a brief scene title of the video segment as the query, requiring the model to simultaneously respond with the precise start and end times of the video segment and provide a detailed description of that segment. While the content in the scene title may leak into the detailed caption response, most of the missing detailed information must be correctly described by attending the corresponding segment. Moreover, temporal grounding and detailed captioning can serve as regularization task for each other, preventing caption model from hallucinations from unrelated visual or linguistic contexts and helping grounding model to regress the timestamp more accurately.

The process for collecting our Temporal Grounded Caption data is described at the bottom of Figure 3(b). In the first stage, we use a detailed caption dataset with timestamps as our data source. We remove data with target grounding time intervals that are too short or too long and ensure that the scenes in the video are as diverse as possible. In the second stage, we use a LLM to summarize scene titles. To prevent excessive semantics of video segments from being leaked from the query to the MLLM, we try to retain the minimal subset of key features that are sufficient to distinguish the video segments. In the third stage, to avoid overly similar or identical content appearing at different temporal intervals in the video, we perform similarity filtering on the data annotations. Based on the scene titles and video features, we calculate the similarity between different segments of the same video and remove data with excessively high similarity. In the fourth stage, we randomly sample the generated data and manually assess its quality. Based on human feedback, we refine the threshold parameters for data filtering used in the first three stages to yield the final Temporal Grounded Caption dataset. This new dataset plays an important role in our grounded tuning.

| Method | LLM Size | Charades-STA | | | QVHighlight | |
|---|---|---|---|---|---|---|
| | | $R@1_{(IOU=0.3)}$ | $R@1_{(IOU=0.5)}$ | $R@1_{(IOU=0.7)}$ | $mAP$ | $HIT@1$ |
| MovieChat (Song et al., 2024a) | 7B | 8.8 | 2.9 | 1.3 | 11.7 | 16.1 |
| GroundingGPT (Li et al., 2024c) | 7B | - | 29.6 | 11.9 | - | - |
| VTimeLLM (Huang et al., 2024a) | 7B | 51.0 | 27.5 | 11.4 | - | - |
| HawkEye (Wang et al., 2024f) | 7B | 50.6 | 31.4 | 14.5 | - | - |
| TimeChat (Ren et al., 2024) | 7B | - | 32.2 | 13.4 | 14.5 | 23.9 |
| ChatVTG (Qu et al., 2024) | 7B | 52.7 | 33.0 | 15.9 | - | - |
| VideoChat2 (Li et al., 2024b) | 7B | 9.6 | 3.4 | 1.4 | 13.4 | 18.6 |
| **VideoChat-T** | **7B** | **69.9** (+60.3) | **48.7** (+45.3) | **24.0** (+22.6) | **26.5** (+13.1) | **54.1** (+35.5) |
| QD-DETR※ (FT) (Moon et al., 2023b) | - | - | 57.3 | 32.6 | 38.9 | 64.2 |
| UnLoc-L※ (FT) (Yan et al., 2023) | - | - | 60.8 | 38.4 | - | - |
| HawkEye (FT) (Wang et al., 2024f) | 7B | 72.5 | 58.3 | 28.8 | - | - |
| Timechat (FT) (Ren et al., 2024) | 7B | - | 46.7 | 23.7 | 21.7 | 37.9 |
| **VideoChat-T (FT)** | **7B** | **79.4** | **67.1** | **43.0** | **27.0** | **55.3** |

Table 1: **Performance of VideoChat-T on temporal grounding and highlight detection tasks.** (FT) indicates the model fine-tuned on training set of the evaluation benchmark, with the respective text marked in gray. Classic supervised expert models are marked with ※.

## 4 EXPERIMENTS

### 4.1 IMPLEMENTATION DETAILS

Built upon VideoChat2, we use UMT-L (Li et al., 2023c) and Mistral-7B (Jiang et al., 2023) as the video encoder and LLM, respectively. Except for the TAPE, all components are initialized from the pre-trained model of VideoChat2-Mistral. For the TAPE, we use random initialization, set the initial values of the final linear layer to zero, and freeze it during the first epoch of training. We set the frame count $T$ for each clip to 8, so the number of clips $K$ for a long video is equal to the total frame count divided by $T$. We fine-tune the model for 3 epochs using the TimePro with 349K instances and a general QA task dataset with 82K instances. To ensure the stability of model training, we use 192-frame input for the first epoch. In the second and third epochs, we unfreeze the TAPE and adjust the model input to 128 frames. All experiments are conducted on 16 A100 GPUs.

### 4.2 PERFORMANCE ON TEMPORAL GROUNDING

We evaluate our method using two commonly used temporal localization tasks, i.e., Temporal Grounding and Highlight Detection. The performance comparison between VideoChat-T and other models is shown in Table 1. Our method's zero-shot performance surpasses all previous LLM-based methods and after fine-tuning, VideoChat-T even exceeds some classic expert models on the temporal grounding task.

**Temporal Grounding.** This task aims to identify the start and end timestamps of the video content described by the query sentence, using Charades-STA as the evaluation benchmark. VideoChat-T achieves an accuracy of 48.7 in the R@1 (IOU=0.5) metric, significantly surpassing the previous state-of-the-art MLLM method, namely TimeChat, by 16.5 points. Additionally, it outperforms the fine-tuned version of TimeChat on the training set of the evaluation benchmark by 2.0%. Furthermore, the performance of VideoChat-T fine-tuned on the evaluation benchmark training set reaches 67.1 R@1 at IoU=0.5, surpassing most state-of-the-art classic supervised expert models.

**Highlight Detection.** We use QVHighlights as the evaluation benchmark. For a given query, this task requires outputting all timestamps of highlight moments and their corresponding saliency scores. Since there could be many sparse highlight moments in a video, this task requires finer-grained video understanding at the frame level. VideoChat-T achieves mAP of 26.5, significantly surpassing the previous MLLM method of TimeChat by 13.0 points, and also outperforms its fine-tuned version by 4.8 points. We observe that after fine-tuning on the corresponding training set, VideoChat-T shows almost no performance improvement. This may be due to the bottleneck in language representation of LLMs. The Highlight Detection task requires outputting a (timestamp, saliency score) pair for each highlight moment, and a video may contain dozens of discrete highlight moments, making it challenging for the model to correctly respond with dozens to hundreds of numbers in a language format. The precise numerical salience score output is very difficult for LLMs, and VideoChat-T can only respond well to queries with fewer highlight moments. Due to the

| Method | LLM Size | Long Video | | | | Short Video |
| | | Egoschema | | VideoMME | | MVbench |
| | | Subset | Full | w/o subs | w/o subs (Long) | Avg |
|---|---|---|---|---|---|---|
| VideoAgent (Wang et al., 2024c) | GPT-4 | 60.2 | 54.1 | - | - | - |
| VideoAgent (Fan et al., 2024) | GPT-4 | 62.8 | - | - | - | - |
| TimeChat (Ren et al., 2024) | 7B | - | 33.0 | 30.2 | 26.1 | 38.5 |
| LLAMA-Vid (Li et al., 2023d) | 7B | - | 38.5 | - | - | 41.9 |
| MovieChat (Song et al., 2024a) | 7B | - | 53.5 | 38.2 | 33.4 | 55.1 |
| MovieChat+ (Song et al., 2024b) | 7B | - | 56.4 | - | - | - |
| Chat-UniVi (Jin et al., 2024) | 7B | - | - | 40.6 | 35.8 | - |
| VideoChat2 (Li et al., 2024b) | 7B | 63.6 | 54.4 | 39.5 | 33.2 | 60.4 |
| **VideoChat-T** | **7B** | **68.4** (+4.8) | **60.0** (+5.6) | **46.3** (+6.8) | **41.9** (+8.7) | **59.9** (-0.5) |

Table 2: **Performance of VideoChat-T and other methods on video question answering tasks.** By upgrading VideoChat2 with TimeSuite, VideoChat-T demonstrates significant improvements across multiple long video benchmarks.

specific architectural design, classic supervised expert models have a natural advantage in handling such tasks, and VideoChat-T still has a performance gap compared to expert models.

### 4.3 PERFORMANCE ON GENERAL VIDEO QA

In addition to test the grounding ability of our VideoChat-T, we also want to verify its general video question answering performance. According to mainstream evaluation standards, we use both long video and short video QA to assess the general video understanding capability of VideoChat-T. Table 2 shows the performance of VideoChat-T on the video QA evaluation benchmarks.

**Long Video QA.** We use Egoschema (Mangalam et al., 2023) and VideoMME (Fu et al., 2024) to evaluate the long video capabilities of VideoChat-T. In conjunction with our proposed architectural improvements, we incremental fine-tune VideoChat2 using only 432K data points. VideoChat-T demonstrates outstanding performance on the Egoschema, achieving an accuracy of 68.4% on the test subset and 60.0% on the entire test set. Compared to VideoChat2, VideoChat-T obtains improvements of 4.8% and 5.6% on the subset and the full test set, respectively. Additionally, for the VideoMME benchmark, VideoChat-T achieves an accuracy of 46.3% by solely analyzing the visual content without using subtitles, representing a 6.8% improvement over VideoChat2. On the long video data division of VideoMME, VideoChat-T achieves an accuracy of 41.9%, which is an 8.7% improvement compared to VideoChat2. The upgraded VideoChat-T demonstrated significant performance improvements on long video QA benchmarks. This indicates the potential of leveraging grounding-centric video tasks to enhance the temporal awareness of MLLMs, thereby further improving long video understanding capabilities.

**Short Video QA.** We use MVBench (Li et al., 2024b) to evaluate the general short video understanding capabilities of VideoChat-T. VideoChat-T achieves an overall average accuracy of 59.9% on MVBench, which is a 0.5% decrease compared to VideoChat2. It is important to note that achieving minimal performance loss is a challenging task. According to previous experiences in the field of incremental learning (Van de Ven et al., 2022), models inevitably forget old knowledge while learning new knowledge. VideoChat2 is fine-tuned with 2M data, whereas VideoChat-T is fine-tuned with only 432K data, where 349K annotations are temporal grounding centric, resulting in only a 0.5% accuracy loss. Previous temporal MLLMs like TimeChat (Ren et al., 2024), although achieving strong temporal localization capabilities, yield much weaker general video QA capability, with an accuracy of only 38.5% on MVBench. This demonstrates that the design of our TimeSuite enhances new capabilities for the model while still preserving the original general video understanding capabilities. For a detailed analysis of the performance degradation of MVBench, please refer to Appendix F.2.

### 4.4 QUALITATIVE ANALYSIS

Figure 4 presents a qualitative comparison between our model and other methods. In the example on the left, VideoChat-T is capable of answering more complex long video reasoning questions. Our model accurately identifies the temporal location of the "light a cigarette" event and determines the correct key clue "the person in a white coat" based on the video content. This leads to the inference

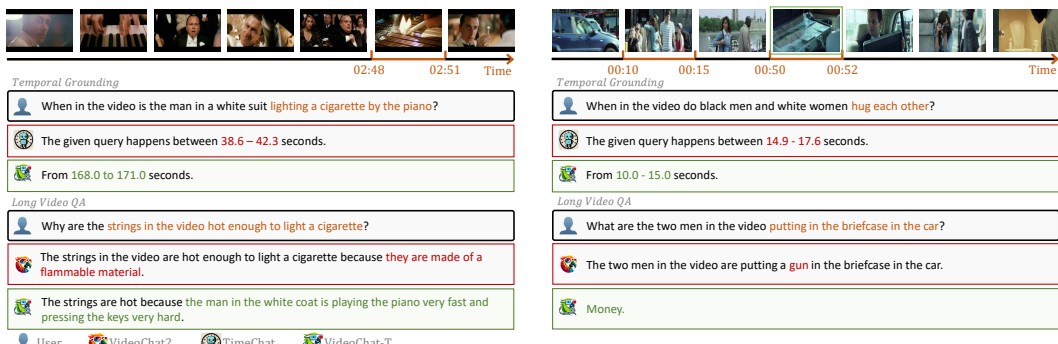

Figure 4: Qualitative comparison between VideoChat-T and other methods. VideoChat-T not only possesses temporal fine-grained perception capabilities but also can perform accurate long video reasoning. Green text indicates correct answers, while red text indicates inappropriate answers.

| Model | Egoschema | VideoMME | Charades-STA | QVHighlights |
|---|---|---|---|---|
| | Full | w/o subs | R@1 IOU=0.5 | Hit@1 |
| VideoChat-T (Ours) | **60.0** | **46.3** | 48.7 | **54.1** |
| w/o TAPE | 59.1 | 45.9 | 47.1 | 50.4 |
| w/o frz | 59.0 | 45.2 | **52.4** | 53.7 |

Table 3: Performance results of the ablation study on the TAPE. Here, w/o adapter refers to removing our proposed TAPE, and w/o frz refers to not using the training method where the TAPE is frozen during the first epoch.

| Model | Egoschema | VideoMME | Charades-STA | QVHighlights |
|---|---|---|---|---|
| | Full | w/o subs | R@1 IOU=0.5 | Hit@1 |
| VideoChat-T(Ours) | **60.0** | **46.3** | **48.7** | **54.1** |
| r/w pooling | 59.8 | 44.8 | 40.3 | 47.3 |
| r/w cluetering | 59.5 | 45.0 | 39.8 | 40.1 |
| w/o init | 57.4 | 43.4 | 42.0 | 53.9 |

Table 4: Performance results of the ablation study on the Token Shuffle. Here, r/w refers to replacing Token Shuffle with the other component, and w/o init refers to removing the efficient initialization.

that "playing the piano very fast and pressing the keys very hard" are the true reasons. The example on the right demonstrates our model's fine-grained perception ability. The appearance of "money in the briefcase" is very brief, and most models easily overlook this detail. Thanks to its strong fine-grained perception ability, our model precisely captures this visual content.

## 4.5 ABLATION STUDY

**Role of TAPE.** To verify the performance improvement brought by TAPE, ablation experiments were conducted. Table 3 lists the performance results of the conducted adapter-related ablation experiments. It can be observed that when the TAPE is removed, the model's performance on long video understanding and temporal grounding benchmarks decreases. TAPE can adaptively embed positional encodings into video tokens, and the absence of TAPE leads to a certain loss in temporal awareness capability. When we unfroze the TAPE in the first epoch, the performance improved on the temporal grounding task but declined on the long video QA task. This is because the TAPE is highly suited for tasks with strong temporal dependencies. If unfrozen too early, the model may become biased towards fitting temporal grounding tasks. Freezing the TAPE during the first epoch allows the model to first optimize and learn a relatively generalized feature representation, thereby balancing the performance across different tasks.

**Effectiveness of Token Shuffle.** To verify the effectiveness of token shuffle, we conducted ablation experiments. Table 4 presents the results of these ablation experiments. We compared token shuffle with conventional methods such as pooling and clustering, and also observed the results after removing efficient initialization. When we replaced token shuffle with pooling or clustering methods, the model's performance declined. This is because the efficient initialization of the linear layer in token shuffle makes the initial values of the module equivalent to average pooling, which gradually optimizes better solutions during training. Therefore, our method is inherently superior to pooling. On the other hand, clustering often fails to maintain the spatial/temporal consistency of the video, leading to temporal confusion. When we removed the efficient initialization of the linear layer, the negative impact of random initialization severely damaged the model's original performance.

**Effect of TimePro.** We conducted ablation studies to evaluate the effectiveness of the TimePro data components. As shown in Table 5, by gradually adding subsets of TimePro, we observed the model's performance changes across various temporal grounding-centric instruction-tuning data. As we pro-

| Normal | TimeIT | TGC | HD | MTG | RTL | Egoschema Full | VideoMME w/o subs | Charades-STA R@1 IOU=0.5 | QVHighlights Hit@1 |
|:---:|:---:|:---:|:---:|:---:|:---:|:---:|:---:|:---:|:---:|
| ✓ | | | | | | 56.6 | 42.6 | 8.0 | 24.4 |
| ✓ | ✓ | | | | | 57.8 | 43.6 | 32.2 | 25.2 |
| ✓ | ✓ | ✓ | | | | 58.3 | 44.0 | 39.1 | 33.9 |
| ✓ | ✓ | ✓ | ✓ | | | 59.8 | 44.9 | 41.9 | 43.8 |
| ✓ | ✓ | ✓ | ✓ | ✓ | | **60.0** | 45.1 | 45.8 | 48.3 |
| ✓ | ✓ | ✓ | ✓ | ✓ | ✓ | **60.0** | **46.3** | **48.7** | **54.1** |

Table 5: Performance results of the ablation study on different components of TimePro. We use 82K normal training data as the baseline. TimeIT refers to the training data with five task types from Ren et al. (2024), TGC refers to Temporal Grounded Caption, HD refers to Highlight Detection, MTG refers to Multi-format Temporal Grounding, and RTL refers to Reasoning Temporal Localization.

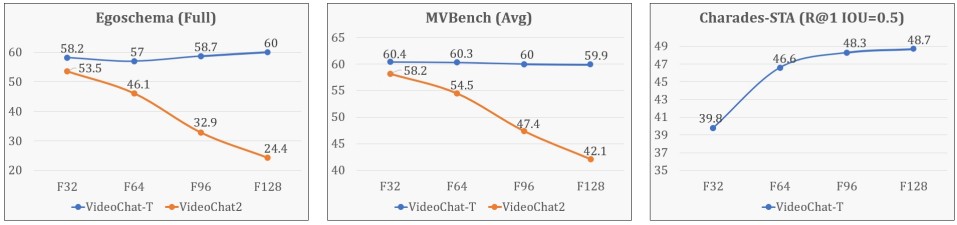

Figure 5: Performance of VideoChat-T with varying input frame numbers. As the number of input frames increases, the performance of VideoChat-T shows an upward trend in both long video QA and temporal grounding tasks. Due to the over low temporal grounding performance of VideoChat2, its curve is omitted.

gressively added subsets of TimePro, not only did the model's performance on temporal grounding tasks show a stable and significant improvement, but we also observed a noticeable upward trend in performance on long video benchmarks. This to some extent corroborates that temporal grounding centric tasks have a positive impact on long video understanding.

**Impact of frames.** To investigate the impact of input frame count on model performance, we conducted an ablation study. Figure 5 illustrates the scalability of our model's performance with respect to input frame count. VideoChat-T demonstrates good stability as the input frame count varies, and its performance in long video QA and temporal grounding tasks improves with an increase in frame count. In contrast, the baseline model, VideoChat2, exhibited catastrophic performance degradation when the frame count was significantly increased. As the input frame count increases, the number of visual encoding tokens grows linearly. Excessive visual token input imposes an additional computational burden on the temporal modeling of the LLM. TimeSuite mitigates this by employing Token Shuffle to reduce the number of tokens, ensuring the stable operation of the model.

## 5 CONCLUSION

In this paper, we have introduced TimeSuite, a collection of new designs from perspectives of efficient architecture, high-quality data, and new instruction tuning task, to achieve long video understanding by fine-tuning short video MLLMs with temporal grounding-centric data. We address the computational challenges of processing long videos by introducing token shuffle to compress visual tokens. We also propose the TAPE for adaptive position encoding, enhancing the temporal awareness of visual representation. Additionally, our designed Temporal Grounded Caption training task ensure MLLMs to build correspondence between grounded segments and detailed caption, while the TimePro dataset provide comprehensive instruction tuning data for learning more effective temporal perception capability. Experimental results demonstrate that VideoChat-T significantly improves long video understanding, with notable performance gains on Egoschema and VideoMME. Furthermore, VideoChat-T exhibits strong zero-shot temporal grounding capabilities, significantly outperforming the previous MLLMs on temporal grounding. Overall, our TimeSuite provides effective designs for short MLLMs to enhance their performance on temporal grounding and long video QA. We hope our TimeSuite could yield some insights on designing long video MLLMs.

ACKNOWLEDGEMENT

This work is supported by the National Key R&D Program of China (No. 2022ZD0160900), the Fundamental Research Funds for the Central Universities (No. 020214380119), Jiangsu Frontier Technology Research and Development Program (No. BF2024076), and the Collaborative Innovation Center of Novel Software Technology and Industrialization.

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

## A  IMPLEMENTATION OF TAPE

**Algorithm 1** PyTorch snippet of TAPE.

Initialize related package

```python
class TemporalAdapter(nn.Module):
    def __init__(self, merge_len, clip_num, input_dim, mid_dim, output_dim, sample_rate):
        super().__init__()
        self.AvgPool = nn.AvgPool1d(merge_len, stride = merge_len)
        self.upsample = nn.Upsample(scale_factor = sample_rate)
        self.linear_input = nn.Linear(input_dim, mid_dim)
        self.linear_output = nn.Linear(mid_dim, output_dim)
        nn.init.constant_(self.linear_output.weight, 0)
        nn.init.constant_(self.linear_output.bias, 0)
        self.Downsample_Depthwise_Separable_Conv1 = nn.Sequential(
            nn.Conv1d(mid_dim, mid_dim, merge_len*2+1, stride=sample_rate, padding=merge_len, groups=mid_dim),
            nn.Conv1d(mid_dim, mid_dim, 1),
            TransposeLayerNorm(mid_dim),
            nn.GELU(),
        )
        self.Downsample_Depthwise_Separable_Conv2 = nn.Sequential(
            nn.Conv1d(mid_dim, mid_dim, merge_len*2+1, stride=sample_rate, padding=merge_len, groups=mid_dim),
            nn.Conv1d(mid_dim, mid_dim, 1),
            TransposeLayerNorm(mid_dim),
            nn.GELU(),
        )
        self.fc = nn.Sequential(
            nn.Conv1d(mid_dim, mid_dim, clip_num+1, stride=1, padding=clip_num//2),
            TransposeLayerNorm(mid_dim),
            nn.GELU(),
        )
        self.Conv2 = nn.Sequential(
            nn.Conv1d(mid_dim, mid_dim, merge_len+1, stride=1, padding=merge_len//2, groups=mid_dim),
            nn.Conv1d(mid_dim, mid_dim, 1),
            TransposeLayerNorm(mid_dim),
            nn.GELU(),
        )
        self.Conv1 = nn.Sequential(
            nn.Conv1d(mid_dim, mid_dim, merge_len+1, stride=1, padding=merge_len//2, groups=mid_dim),
            nn.Conv1d(mid_dim, mid_dim, 1),
            TransposeLayerNorm(mid_dim),
            nn.GELU(),
        )

    def forward(self, input_tokens):
        time_ad = self.linear_input(input_tokens).transpose(1, 2)
        time_ad1 = self.AvgPool(time_ad)
        time_ad2 = self.Downsample_Depthwise_Separable_Conv1(time_ad1)
        time_ad3 = self.Downsample_Depthwise_Separable_Conv2(time_ad2)
        time_ad3 = self.fc(time_ad3)
        time_ad2 = self.upsample(time_ad3)+ time_ad2
        time_ad2 = self.Conv2(time_ad2)
        time_ad1 = self.upsample(time_ad2)+ time_ad1
        time_ad1 = self.Conv1(time_ad1)
        time_ad_out = self.linear_output(time_ad1.transpose(1, 2))
        return time_ad_out
```

Algorithm 1 details the implementation process of TAPE in code form. Specifically, the long video token sequence $input\_tokens$ is first compressed in the channel dimension by a linear layer to obtain $time\_ad$, and the sequence length is compressed through a pooling layer. Next, we use a U-Net-like structure composed of one-dimensional depthwise separable convolutions to progressively down-sample the sequence, obtaining three one-dimensional temporal feature sequences with different

time resolutions, namely $time\_ad1$, $time\_ad2$, and $time\_ad3$. Subsequently, a convolution with a sufficiently long window is applied to the shortest temporal feature sequence $time\_ad3$, using zero padding at both ends as anchors to encode the relative temporal position of each token in the sequence. Then, we progressively upsample the temporal feature sequences from short to long, using residual connections to preserve temporal features at different scales. Finally, the temporal feature sequence $time\_ad\_out$ is restored to the same length as the video features after token shuffling and aligned in the channel dimension through a linear layer.

## B  INSTRUCTION-TUNING DATA

We fine-tuned VideoChat-T using 432K data, which includes 349K instances from TimePro and 82K instances from normal data. All videos were sampled from existing open-source video datasets, with specific information about the relevant data provided in Table 6.

| Set | Task | Source | Instance Num |
|---|---|---|---|
| TimePro | Temporal Video Grounding | DideMo | 32,944 |
| | | QueryD | 14,602 |
| | | HiREST-grounding | 459 |
| | Dense Video Captioning | ActivityNet-Captions | 10,009 |
| | | ViTT | 5,086 |
| | | Youcook2 | 8,700 |
| | Video Summarization | TVSum | 50 |
| | | SumMe | 25 |
| | Step Localization and Captioning | COIN | 9,026 |
| | | HiREST-step | 459 |
| | Transcribed Speech Generation | YT-Temporal | 31,190 |
| | Reasoning Temporal Localization | ActivityNet-RTL | 33,557 |
| | Multi-format Temporal Grounding | InternVid-VTime | 100,000 |
| | Highlight Detection | ActivityNet-HL | 10,340 |
| | Temporal Grounded Caption | CosMo-TGC | 93,118 |
| Normal | Conversation | VideoChatGPT | 13,303 |
| | | VideoChat | 13,884 |
| | Video QA | EgoQA | 7,813 |
| | | MovieChat-QA | 808 |
| | Reasoning | STAR | 45,731 |
| | Caption | MovieChat-Caption | 808 |

Table 6: The complete instruction fine-tuning data used for training. We utilized a total of approximately 432K data points, which can be divided into 349K instances of TimePro and 82K instances of regular video data, covering 13 tasks across 21 datasets.

We evaluate the quality of the data from three perspectives: diversity, length, and difficulty. We strive to include different datasets for various tasks, and the distribution of videos in the datasets is as broad as possible. The length of the videos should be controlled within an appropriate range, as excessively long or short videos may pose challenges for training. Each query should clearly describe the video content of the target time segment and avoid corresponding to multiple time segments in the video. Based on these principles, we have screened and integrated existing high-quality datasets, which significantly contribute to enhancing the model's temporal awareness capabilities.

TimePro encompasses a series of open-source temporal grounding datasets that we have integrated, cleaned, and refined, such as TimeIT (Ren et al., 2024), ANet-RTL (Huang et al., 2024b), and InternVid-VTime (Huang et al., 2024a). These high-quality open-source datasets have been experimentally validated by us. We also added two new self-made datasets, ANet-HL and CosMo-TGC.

**Temporal Video Grounding.** This task involves providing a natural language query and requires outputting the corresponding video's start and end times. The datasets include DiDeMo (Anne Hen-

dricks et al., 2017), QuerYD (Oncescu et al., 2021), and HiREST-grounding (Zala et al., 2023), aiming to achieve precise temporal localization during user interaction with natural language.

**Dense Video Captioning.** This task requires the model to detect a series of events occurring in a given video and output the corresponding timestamps and coarse-grained descriptions. The datasets for this part include ActivityNet-Caption (Krishna et al., 2017), ViTT Huang et al. (2020), and YouCook2 (Zhou et al., 2018), which help the model learn the temporal relationships between different events within the video.

**Video Summarization.** The goal of this task is not to summarize at the semantic level of natural language, but to determine a set of compressed frames or clips in the form of timestamps, representing the most informative content in a given video. Our datasets include TVSum (Song et al., 2015) and SumMe (Gygli et al., 2014), which effectively combine the model's temporal perception capabilities with its semantic content inference abilities.

**Step Localization and Captioning.** This task differs from dense video captioning as it is designed to segment and describe the important steps within a long video. We have integrated two datasets, COIN (Tang et al., 2019) and HiREST-step (Zala et al., 2023), which can help the model learn the procedural temporal logic relationships of different steps within a single event.

**Transcribed Speech Generation.** The purpose of this task is to predict speech content and its corresponding start and end timestamps based on visual signals in the video. Including the YT-Temporal (Zellers et al., 2022) dataset, this task can be viewed as a weakly supervised event localization and description task.

**Reasoning Temporal Localization.** The answers to the questions in this task include both timestamps and explanations. We used the ANet-RTL (Huang et al., 2024b) dataset as training data for this task. By combining temporal localization and reasoning, we can more specifically enhance the model's temporal perception capabilities.

**Multi-format Temporal Grounding.** This task includes both single-turn and multi-turn dialogues, with a variety of question types. We use the InternVid-VTime (Huang et al., 2024a) dataset for training this task. The broader range of task types and more diverse output formats can effectively enhance the model's temporal generalization capabilities.

**Highlight Detection.** Unlike video summarization, this task identifies only the most salient moments of a video in response to a natural language query, without covering the entire scope of the original video (Lei et al., 2021a). We used a custom dataset, ANet-HL, derived from temporal localization data. We extract video segments between the start and end times of the target's appearance and use CLIP to calculate the similarity between each frame's scene and the target. This is converted into discrete saliency levels ranging from 1 to 5, at intervals of 0.5. This task effectively enhances the model's temporal perception capabilities for specific events.

**Temporal Grounded Caption.** This task involves using scene titles as queries, requiring the model to output both the time segments when the scenes appear and the fine-grained subtitles for those segments. We used our custom dataset, CosMo-TGC. This task format, which combines temporal localization and semantic understanding, can effectively prevent large language models from focusing on irrelevant video segments, thereby improving the quality of the model's responses to questions.

We also used normal data comprising four tasks and six different data sources. These general data help prevent the model from overfitting to temporal grounding-related tasks during training, thereby preserving the model's original capabilities.

## C  COMPUTATIONAL EFFICIENCY

By applying Token Shuffle, we further reduced the computational cost of VideoChat-T, giving it a significant computational advantage over high-performance models like LLaVA-OneVision (Li et al., 2024a) and Qwen2-VL (Wang et al., 2024a). Under the same settings, VideoChat-T uses only 3 tokens per frame, with flops consumption at just 5.1% of LLaVA-OneVision. Its inference time on single A100 is only 0.63 seconds, reaching real-time response levels, making it highly suitable for applications requiring rapid response, such as online video understanding.

| Method | Token num per frame | flops 128 frames | Inference Time 128f & on single A100 GPU | Charades-STA IOU0.5 | QVHighlight mAP | MVBench Avg | Egoschema Full | VideoMME Vision |
|---|---|---|---|---|---|---|---|---|
| Qwen2-VL (Wang et al., 2024a) | 138 | 929.8 T | Out Of Memory | 15.0 | 13.0 | 67.0 | 66.7 | 63.3 |
| LLaVA-OneVision (Li et al., 2024a) | 196 | 693.7 T | 4.95 s | 7.3 | 14.98 | 56.7 | 60.1 | 58.2 |
| VideoChat-T (Ours) | 3 | 35.5 T | 0.63 s | 48.7 | 26.5 | 59.9 | 60.0 | 46.3 |

Table 7: Comparison of the computational efficiency and performance of VideoChat-T with other methods. Our approach achieves relatively impressive performance with extremely low computational cost.

In terms of performance, VideoChat-T significantly outperforms LLaVA-OneVision in temporal grounding tasks. It has a slight advantage on MVBench; both perform comparably on Egoschema; but VideoChat-T performs worse on VideoMME. Given the substantial savings in computational resources with VideoChat-T, we consider the disadvantages on some datasets to be acceptable.

Moreover, our model's ability to maintain reasonable performance under high compression ratios suggests that the token embedding spaces of contemporary models may be characterized by considerable feature redundancy. This observation presents a promising avenue for future research, as efficient techniques for compressing or discarding redundant features could substantially reduce computational costs without sacrificing model performance, enabling longer context reasoning.

## D  DETAILS OF HYPERPARAMETERS

| config | epoch1 | epoch2&3 |
|---|---|---|
| input frame | 192 | 128 |
| max text length | 1536 | 1024 |
| freeze TAPE | True | False |
| learning rate | 2e-5 | 1.5e-5 |
| input resolution | 224 | |
| clip frame | 8 | |
| merge lenth | 4 | |
| QFormer token (per clip) | 96 | |
| lora rank | 16 | |
| lora alpha | 32 | |
| lora dropout | 0.1 | |
| batch size (per GPU) | 2 | |
| optimizer | AdamW | |
| optimizer momentum | 0.9, 0.999 | |
| weight decay | 0.02 | |
| learning rate schedule | cosine decay | |

Table 8: Hyper-parameter Settings During the Training Process of VideoChat-T.

Table 8 lists the hyperparameters used during different epochs of the training process. In the first epoch, we used a larger number of input frames and froze the TAPE. At the beginning of the second epoch, we unfroze the TAPE and fixed the model's input frames to 128. Following the settings of VideoChat2, we integrated the lora module into the LLM and applied flash attention to accelerate the training process.

## E  FULL PERFORMANCES

| Model | LLM | Avg | AS | AP | AA | FA | UA | OE | OI | OS | MD | AL | ST | AC | MC | MA | SC | FP | CO | EN | ER | CI |
|---|---|---|---|---|---|---|---|---|---|---|---|---|---|---|---|---|---|---|---|---|---|---|
| VideoChatGPT (Maaz et al., 2023) | 7B | 32.7 | 23.5 | 26.0 | 62.0 | 22.5 | 26.5 | 54.0 | 28.0 | 40.0 | 23.0 | 20.0 | 31.0 | 30.5 | 25.5 | 39.5 | 48.5 | 29.0 | 33.0 | 29.5 | 26.0 | 35.5 |
| VideoLLaMA (Zhang et al., 2023) | 7B | 34.1 | 27.5 | 25.5 | 51.0 | 29.0 | 39.0 | 48.0 | 40.5 | 38.0 | 22.5 | 22.5 | 43.0 | 34.0 | 22.5 | 32.5 | 45.5 | 32.5 | 40.0 | 30.0 | 21.0 | 37.0 |
| VideoChat (Li et al., 2023b) | 7B | 35.5 | 33.5 | 26.5 | 56.0 | 33.5 | 40.5 | 53.0 | 40.5 | 30.0 | 25.5 | 27.0 | 48.5 | 35.0 | 20.5 | 42.5 | 46.0 | 26.5 | 41.0 | 23.5 | 23.5 | 36.0 |
| ST-LLM (Liu et al., 2024b) | 7B | 54.9 | 66.0 | 53.5 | 84.0 | 44.0 | 58.5 | 80.5 | 73.5 | 38.5 | 42.5 | 31.0 | 86.5 | 36.5 | 56.5 | 78.5 | 43.0 | 44.5 | 46.5 | 34.5 | 41.5 | 58.5 |
| VideoChat2 (Li et al., 2024b) | 7B | 60.4 | 75.5 | 58.0 | 83.5 | 50.5 | 60.5 | 87.5 | 74.5 | 45.0 | 47.5 | 44.0 | 82.5 | 37.0 | 64.5 | 87.5 | 51.0 | 66.5 | 47.0 | 35.0 | 37.0 | 72.5 |
| VideoChat-T | 7B | 59.9 | 83.5 | 68.5 | 80.5 | 44.0 | 61.0 | 71.0 | 84.0 | 35.5 | 48.0 | 56.5 | 87.0 | 46.0 | 56.5 | 78.0 | 49.5 | 59.0 | 46.0 | 37.0 | 40.0 | 66.5 |

Table 9: The full performance of VideoChat-T on MVBench. VideoChat-T still demonstrates strong performance, effectively prevents catastrophic forgetting caused by incremental fine-tuning.

The performance of VideoChat-T on MVBench is shown in Table 9. Compared to VideoChat2, VideoChat-T only experienced a 0.5% accuracy loss. This indicates that our method effectively preserves the capabilities of the base model, preventing catastrophic forgetting caused by incremental fine-tuning. For a detailed analysis of the performance degradation of MVBench, please refer to Appendix F.2. For the Action Localization (AL) task, which requires the model to determine the coarse-grained temporal position of events, the test accuracy improved from 44.0% to 56.5%. This indirectly confirms that our method significantly enhances the model's temporal awareness capabilities.

| Model | LLM size | Overall (%) | | Short Video (%) | | Medium Video (%) | | Long Video (%) | |
|---|---|---|---|---|---|---|---|---|---|
| | | w/o subs | w subs | w/o subs | w subs | w/o subs | w subs | w/o subs | w subs |
| ST-LLM (Liu et al., 2024b) | 7B | 37.9 | 42.3 | 45.7 | 48.4 | 36.8 | 41.4 | 31.3 | 36.9 |
| Video-LLaVA (Lin et al., 2023a) | 7B | 39.9 | 41.6 | 45.3 | 46.1 | 38.0 | 40.7 | 36.2 | 38.1 |
| ShareGPT4Video (Chen et al., 2024) | 8B | 39.9 | 43.6 | 48.3 | 53.6 | 36.3 | 39.3 | 35.0 | 37.9 |
| Chat-UniVi-v1.5 (Jin et al., 2024) | 7B | 40.6 | 45.9 | 45.7 | 51.2 | 40.3 | 44.6 | 35.8 | 41.8 |
| Qwen-VL-Chat (Bai et al., 2023) | 7B | 41.1 | 41.9 | 46.9 | 47.3 | 38.7 | 40.4 | 37.8 | 37.9 |
| ShareGemini (Share, 2024) | 7B | 43.2 | 47.9 | 49.1 | 52.8 | 41.3 | 47.3 | 39.1 | 43.4 |
| VideoChat2 (Li et al., 2024b) | 7B | 39.5 | 43.8 | 48.3 | 52.8 | 37.0 | 39.4 | 33.2 | 39.2 |
| VideoChat-T | 7B | 46.3 | 55.8 | 53.3 | 59.9 | 43.8 | 54.0 | 41.9 | 53.4 |

Table 10: The full performance of VideoChat-T on VideoMME. VideoChat-T achieved significant performance improvements, particularly in the long video subset.

The overall performance of our model on VideoMME is presented in Table 10. VideoChat-T achieved significant improvements on both evaluation benchmarks of VideoMME, which include watching videos only and videos with subtitles. The improvements are particularly notable in the long video subset.

# F EXTRA ABLATION

## F.1 DOMAIN CORRELATION OF DATA

| Model | Charades-STA(R@1 IOU=0.5) | MVBench(avg) |
|---|---|---|
| VideoChat-T | 48.7 | 59.9 |
| w/o STAR | 47.5 (-1.2) | 59.4 (-0.5) |

Table 11: The performance changes of the model after removing STAR. Although the video sources of STAR may have some domain correlation with those of Charades-STA and MVBench, the performance of our model is minimally affected by STAR.

We found that the video sources in the STAR dataset might have some domain correlation with the video sources in MVBench and Charades-STA. Therefore, we removed STAR from the training set while keeping other training settings consistent with the original. The performance on benchmarks where the video sources might have domain correlation is shown in Table 11. The model's accuracy on Charades-STA (R@1 IOU=0.5) decreased by 1.2%, and the average accuracy on MVBench decreased by 0.5%. This indicates that the domain correlation of video sources did not significantly impact performance for our model. Notably, after removing STAR, our normal data volume was reduced to approximately 36K. This implies that, with sufficiently parameter-efficient initialization and appropriate training strategies, using only a small amount of high-quality normal data is sufficient to retain the model's original capabilities.

## F.2 DEEPER INVESTIGATION OF THE PERFORMANCE DROP ON MVBENCH

We conducted a deeper investigation into the performance decline on MVBench. Through additional ablation experiments (as shown in Tabel 12) , we identified two main factors contributing to the performance drop.

Architectural Discrepancy: The original VideoChat2 model was designed to process only 16 frames, leading to a mismatch in the learned feature distribution compared to the architecture of VideoChat-T. As shown in the first two rows of the table, increasing the input frame number for VideoChat2

| Method | post ft data | data size | frame num | token num (per frame) | MVBench(AVG) |
|---|---|---|---|---|---|
| VideoChat2 | - | - | 16 | 12 | 60.4 |
| VideoChat2 | - | - | 128 | 12 | 42.1 |
| VideoChat-T (Common_Init) | - | - | 128 | 3 | 25.3 |
| VideoChat-T (Ours) | - | - | 128 | 3 | 48.6 |
| VideoChat-T (Ours) | TimePro+Normal (Ours) | 0.43M | 128 | 3 | 59.9 |
| VideoChat-T (Ours) | TimePro+FullVideoChat2 | 2M | 128 | 3 | 62.9 |

Table 12: Performance of VideoChat2 and VideoChat-T on MVBench under different settings.

resulted in a significant performance drop (from 60.4 to 42.1). When initializing VideoChat-T with VideoChat2, performance was close to random (25.3) due to the newly introduced randomly initialized layers. By applying efficient initialization to these new layers, we partially recovered the original capabilities of the model, bringing the MVBench performance of the un-trained VideoChat-T back to 48.6, representing an improvement of 6.5 compared to the 128-frame VideoChat2. After further fine-tuning, the short-video processing capability of VideoChat-T improved significantly, reaching 59.9.

Fine-tuning Data Discrepancy: We fine-tuned VideoChat-T using only 432K data, significantly less than the 2M non-grounded regular data used for training VideoChat2. The fine-tuning data for VideoChat2 primarily consisted of short videos of around ten seconds, which closely matched the length distribution of the MVBench evaluation videos, playing a crucial role in improving MVBench performance. To validate our hypothesis, we conducted additional experiments by training our VideoChat-T model using the TimePro and full VideoChat2 training data. It can be observed that VideoChat-T showed a slight improvement in performance on the MVBench dataset, achieving an accuracy of 62.9, which is an increase of 2.5 compared to the original VideoChat2.

Based on the above, we can conclude the fundamental reasons affecting the model's foundational generalization capabilities. When a model undergoes adjustments, the learned original distribution may not perfectly match the new architecture, making the efficient initialization of new layers crucial. The features learned from the original dataset might be forgotten due to changes in various parameters. Utilizing a more comprehensive and diverse dataset for fine-tuning can restore and even further enhance performance.

### F.3 ASSOCIATION BETWEEN PERFORMANCE AND MODEL DESIGN

| Method | FT Data | Charades-STA IOU0.5 | QVHighlight mAP | MVBench Avg | Egoschema Full | VideoMME w/o subs |
|---|---|---|---|---|---|---|
| TimeChat | TimeIT+Valley | 32.2 | 14.5 | 38.5 | 33.0 | 30.2 |
| TimeChat | TimePro+Normal | 34.2 | 16.3 | 41.6 | 38.9 | 33.4 |
| VideoChat-T | TimePro+Normal | 48.7 | 26.5 | 59.9 | 60.0 | 46.3 |

Table 13: Comparison of other model architectures trained on our dataset with our method, demonstrating the impact of the overall model structure design.

To eliminate the influence of training data and auxiliary tasks, and to more clearly evaluate the association between performance and model design, we fine-tuned TimeChat using the full set of fine-tuning data and auxiliary tasks from VideoChat-T. Table 13 presents the performance of TimeChat, fine-tuned with our data, across five datasets. It can be observed that TimeChat, fine-tuned with our data, shows improvements across all benchmarks. However, its performance still lags significantly behind VideoChat-T. This indicates that an efficient fine-tuning architecture design and high-quality, diverse datasets are both essential and complementary.

### F.4 VALIDATION OF TRANSFERABILITY

To verify the robustness of our TimeSuite for other MLLMs, we transferred our method to Llava-OneVision (Li et al., 2024a). Table 14 shows the performance changes of Llava-OneVision after applying our TimeSuite. It can be seen that when we apply the full set of methods in TimeSuite to Llava-OneVision, the model's performance on two different long-video evaluation benchmarks

| Method | Charades-STA IOU0.5 | QVHighlight mAP | VideoMME w/o subs | MLVU Avg | MVBench Avg |
|---|---|---|---|---|---|
| Llava-OneVision (baseline) | 7.3 | 15.0 | 58.2 | 64.7 | 56.7 |
| Llava-OneVision-T (Ours) | 42.5 | 21.7 | 61.4 | 69.4 | 56.1 |

Table 14: Performance comparison of TimeSuite migration to other MLLMs. The application of our method shows a certain improvement in long video comprehension, demonstrating the transferability of our approach.

improves (+3.2 on VideoMME and +4.7 on MLVU), effectively demonstrating the robustness of our TimeSuite for different MLLMs.

### F.5 EXPLORATIONS OF DATA CONFIGRATIONS OF TIMEPRO

| Method | MVBench Avg | Egoschema Full | VideoMME w/o subs | Charades-STA IOU=0.5 | QVHighlight mAP |
|---|---|---|---|---|---|
| TimePro615K+Normal82K (old version) | 60.0 | 61.0 | 46.3 | 45.4 | 25.7 |
| TimePro349K+Normal82K (Ours) | 59.9 | 60.0 | 46.3 | 48.7 | 26.5 |

Table 15: Comparison of different versions of our proposed TimePro. More data does not necessarily lead to higher overall performance, highlighting the importance of data quality.

In the early version of TimePro, we employed datasets comprising 309K Multi-format Temporal Grounding instances, 150K Temporal Grounded Caption instances and other data. Through extensive experimentation (as shown in Tabel 15), we discovered that removing low-quality data while retaining high-quality instances could significantly reduce training time without compromising performance. Consequently, we pruned these two part datasets to 100K and 93K instances, respectively. The data distribution presented in the paper represents the optimized and relatively balanced configuration we arrived at.

## G  DISCUSSION

### G.1  CAN THE OVERALL PERFORMANCE OF MLLMs BE ENHANCED BY CONTINUOUSLY INTEGRATING EXPERT TASKS?

By appropriately fine-tuning the Multimodal Large Language Model (MLLM), we have developed a general MLLM with powerful zero-shot temporal grounding capabilities. Its performance, after fine-tuning on the training set of evaluation benchmarks, can rival the current state-of-the-art supervised expert models. Based on these results, we can boldly speculate whether it is possible to internalize the capabilities of expert models such as spatial grounding, tracking and detection (Zeng et al., 2023) into the MLLM itself, without using any external expert decoders, to enhance the comprehensive understanding performance of the MLLM and achieve a unified generalist MLLM for multiple tasks.

Merlin (Yu et al., 2023) and VisionLLM (Wang et al., 2024b) have already attempted something similar, but its performance is limited by the reasoning capabilities and language representation bottlenecks of the LLM. There is still a significant gap between its performance and that of expert models for various tasks. We observed similar phenomena in our experiments. The temporal grounding task only requires outputting two timestamps, and the task format is relatively simple, so our model achieved good results. However, the highlight detection task requires outputting multiple discrete timestamps and their corresponding saliency scores. The model needs to accurately predict dozens of numbers in language form to answer the question correctly. Our model performed well only on data with fewer timestamps. Therefore, how to simplify the complex output format of expert tasks into the language representation of LLMs, or to design special processing procedures to simplify complex expert tasks, is a question worth exploring.

Moreover, designing diverse data formats is also crucial for enhancing the expert capabilities of MLLMs. Compared to classic expert models, MLLMs have a natural advantage in task type diversity and can enhance their performance through various different variants tasks of a single capability.

For temporal grounding tasks, we found that enhancing task diversity has a significant effect on improving the model's temporal perception generalization ability. We can boldly speculate that if there are sufficiently diverse training data task types, most tasks with relatively simple output formats can achieve results comparable to expert models through appropriate instruction fine-tuning.

Through the integration of diverse expert tasks and the optimization of language representations, MLLMs can achieve substantial improvements in their overall capabilities. This allows them to effectively comprehend and address complex tasks, rivaling or even exceeding the performance of specialized expert models within specific domains. Looking ahead, MLLMs have the potential to evolve into highly versatile AI models, transcending traditional conversational and QA capabilities. They will be equipped to handle a wide range of complex expert tasks across various domains, such as vision, language, and reasoning.

### G.2    WHY DOES TEMPORAL GROUNDING DATA LEAD TO ACCURACY LOSS IN SHORT-TERM VIDEOS?

We conducted ablation experiments using different combinations of temporal grounding data and regular data. The accuracy of VideoChat-T on MVBench after fine-tuning with various data combinations is shown in Table 16.

| FT Data | MVBench (AVG) |
|---|---|
| TimeIT | 54.7 |
| TimeIT+Normal | 55.3 |
| Normal | 56.1 |
| TimePro | 57.4 |
| TimePro+Normal (Ours) | 59.9 |

Table 16: Performance VideoChat-T on MVBench under different fine-tuning data settings.

The diversity of grounding data formats in the past has often been limited, which can lead to over-fitting on Temporal Grounding tasks and cause the model to lose its general question-answering capability. We compared the TimeIT dataset proposed in TimeChat (Ren et al., 2024) with our TimePro dataset on MVBench. As shown in the Table 16, fine-tuning with only TimeIT resulted in the lowest accuracy, and the combined use of TimeIT+Normal also performed slightly worse than using Normal alone. This indicates that monotonous grounding data indeed damages the model's original performance (as shown in Figure 1 at the beginning of the paper, TimeChat loses some of its general question-answering capability after fine-tuning, where it outputs localization times for general questions).

In contrast, our TimePro dataset includes diverse data, encompassing 9 different task types from 15 datasets, which helps mitigate the generalization loss caused by homogeneous grounding data types. Additionally, our dataset integrates Grounding with various general tasks. For instance, Grounded Caption requires detailed descriptions of corresponding video segments, while Reasoning Temporal Localization demands the model to reason about questions. This approach significantly enhances the model's generalization ability and minimizes the impact on its original capability (e.g., short video accuracy). As demonstrated in the Table 16, the performance of using only TimePro exceeds that of using Normal alone, and the combined use of TimePro and Normal far surpasses all other combinations. This also confirms that our TimePro effectively preserves the model's original performance.

Overall, using a single type of expert task training data can easily lead to model overfitting, resulting in significant loss of the model's original capabilities. To preserve the model's foundational generalization abilities, it is essential to use diversified training data. Additionally, incorporating data of various types and distributions, such as text, images, and videos, can further enhance the model's generalization capabilities.

### G.3 COULD TRAINING THE MODEL ON BOTH TEMPORAL AND NON-TEMPORAL GROUNDING DATA MITIGATE PERFORMANCE LOSS IN SHORT-TERM VIDEOS?

To address this question, we conducted additional ablation experiments. By training VideoChat-T with different combinations of temporal and non-temporal grounding data, we were able to clearly observe the effects of both types of data on the model's performance. The results of the experiments are shown in the Table 17.

| FT Data | MVBench Avg | VideoMME w/o subs | Charades-STA R1@0.5 |
|---|---|---|---|
| Normal | 56.1 | 42.6 | 8.0 |
| TimePro | 57.4 | 46.0 | 45.6 |
| TimePro+Normal (Ours) | 59.9 | 46.3 | 48.7 |

Table 17: Performance comparison of VideoChat-T using different combinations of temporal grounding and non-temporal grounding data.

It can be observed that the combined use of TimePro+Normal for VideoChat-T achieves the highest performance in short video QA, long video QA, and temporal grounding tasks. This not only demonstrates that using both temporal grounding and non-temporal grounding data can reduce performance loss in short videos, but also reveals that the effects of temporal and non-temporal grounding data are complementary across various tasks. The distinct differences between temporal grounding and non-temporal grounding tasks can respectively compensate for the model's shortcomings in different task perspectives and feature distributions. The simultaneous use of both types of data can effectively enhance the model's overall capabilities.

## H CASE STUDY

### H.1 MORE QUALITATIVE ANALYSIS

To further qualitatively analyze our model, we supplemented it with three types of examples. These examples are about long video QA, short video QA, and captioning tasks, all of which include temporal grounding.

More qualitative comparisons about long video QA are shown in Figure 6. VideoChat-T effectively handles various questions across different domains. By better perceiving the temporal relationships of different events occurring in long videos, it can more accurately and deeply understand the detailed content of the entire video.

More qualitative comparisons about short video QA are shown in Figure 7. VideoChat-T effectively retains the original capabilities of the base model. Through parameter-efficient initialization methods and appropriate training strategies, we minimize the damage to the base model's capabilities caused by new architectures and data.

More qualitative comparisons about captioning are shown in Figure 8. Although VideoChat2 describes more local details in some scenarios compared to VideoChat-T, VideoChat-T focuses more on a series of temporal events, which aligns better with how humans typically describe videos.

### H.2 SHORTCOMINGS

We also conducted a qualitative analysis of the shortcomings of VideoChat-T through examples. As shown in Figure 9, VideoChat-T performs poorly on examples with complex logic. In the left example, although VideoChat-T accurately identified the timing of the event, it failed to fully explain the motivation behind the man opening the isolation door, which was "to fight the hijackers of the space elevator, seize the controller, and thus save the people in the entire space elevator." In the right example, VideoChat-T correctly identified the event where Mr. Bean reached out to touch his desk mate's table, but it incorrectly explained the true reason for this action, which was "to cover up the fact that he was copying his desk mate's exam by pretending to wipe dust off the desk."

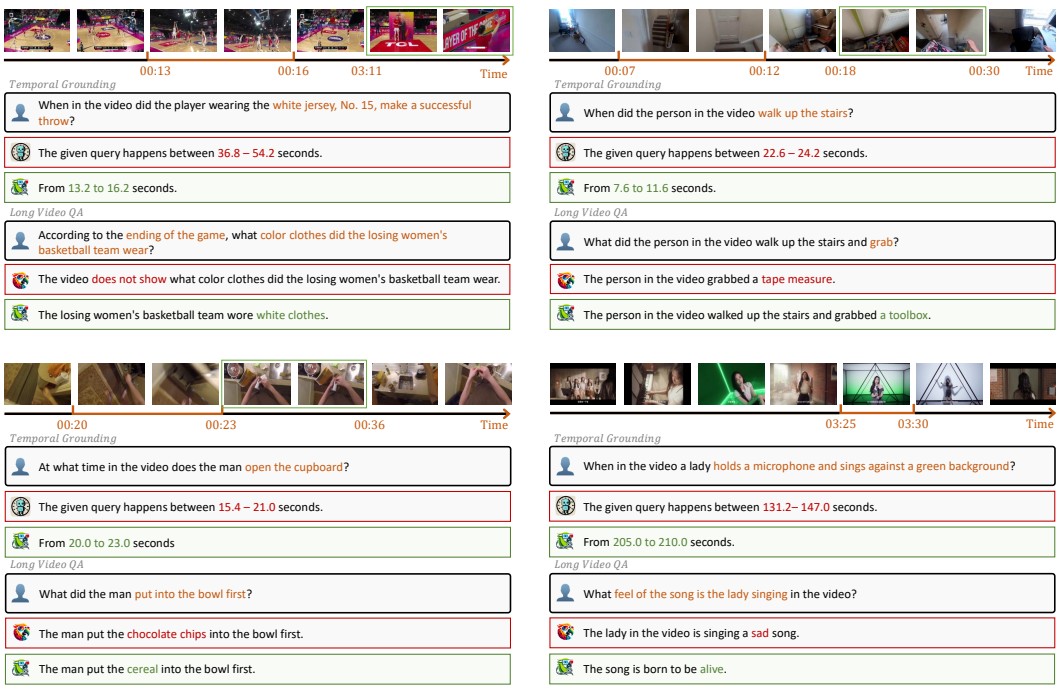

Figure 6: More qualitative comparisons in temporal grounding & long video QA.

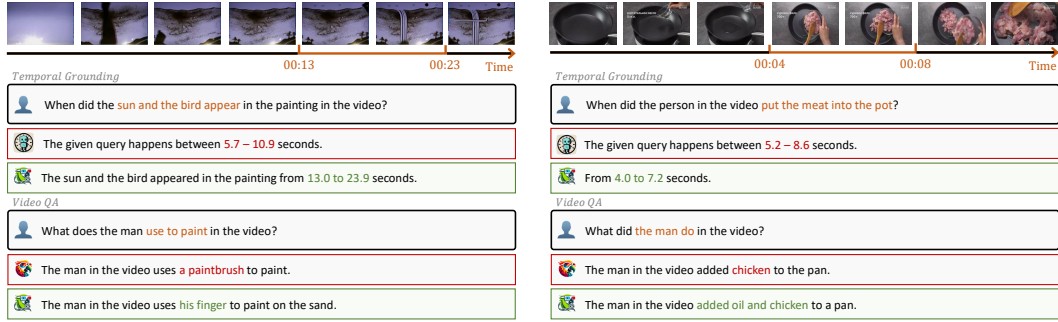

Figure 7: More qualitative comparisons in temporal grounding & short video QA.

Due to the preponderance of single-turn, perceptual questions in our training data and the lack of multi-step reasoning data with complex logic, our model struggles to handle more challenging scenarios that demand intricate logical reasoning. To address this limitation, we propose constructing data in a chain-of-thought format to guide the model through multi-step reasoning, enabling it to delve deeper into the underlying motivations and causal relationships within a video.

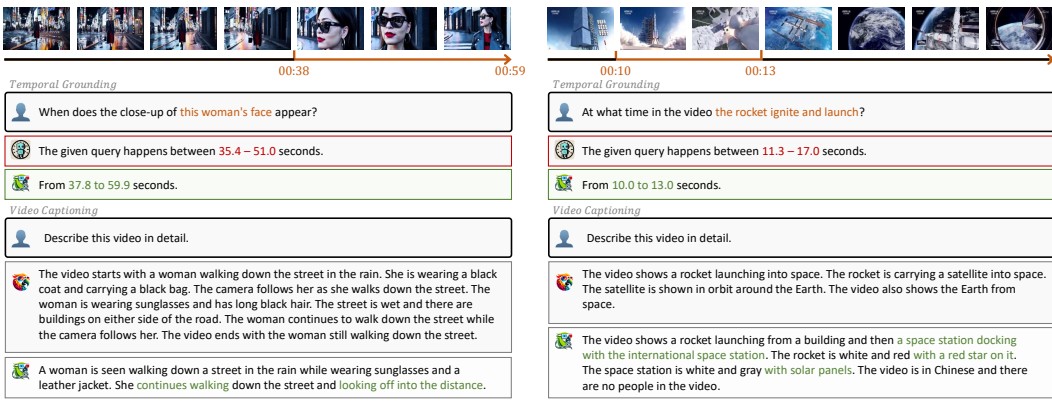

Figure 8: More qualitative comparisons in temporal grounding & captioning.

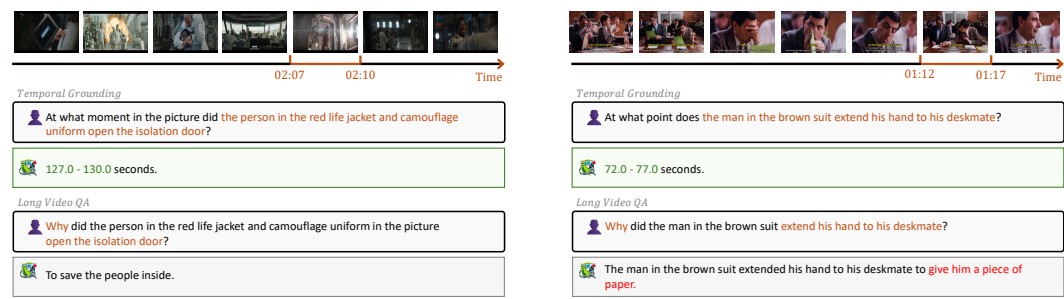

Figure 9: Examples of poor performance by VideoChat-T. While it accurately identifies the time of events, it struggles to answer questions that involve more complex logic.

