# OpenReview forum: "TimeSuite: Improving MLLMs for Long Video Understanding via Grounded Tuning"
_ICLR.cc/2025/Conference — ICLR 2025 Poster_

### Official Review · Reviewer_epk2 · 2024-10-28

**Soundness:** 2
**Presentation:** 2
**Contribution:** 2
**Rating:** 8
**Confidence:** 4

**Summary:**

The paper introduces TimeSuite, a novel framework designed to enhance the capabilities of Multimodal Large Language Models (MLLMs) for understanding long videos through grounded tuning. TimeSuite comprises three key components: a token shuffling scheme to compress visual tokens, reducing computational load; Temporal Adaptive Position Encoding (TAPE) to boost the temporal sensitivity of visual representations; and a new instruction tuning task, Temporal Grounded Caption, which integrates timestamp prediction to guide MLLMs to focus on relevant visual content. The authors also present TimePro, a high-quality dataset with 349k grounded annotations across 9 tasks, aimed at improving MLLMs' temporal perception. Experiments demonstrate that TimeSuite significantly improves performance on long video understanding benchmarks like Egoschema and VideoMME, with improvements of 5.6% and 6.8% respectively, and exhibits robust zero-shot temporal grounding abilities, outperforming existing MLLMs. The paper concludes that TimeSuite provides effective designs for MLLMs to enhance their performance on temporal grounding and long video question answering.

**Strengths:**

- **Originality**: The paper introduces TimeSuite, a novel framework that enhances Multimodal Large Language Models (MLLMs) for long video understanding through grounded tuning. It proposes creative solutions like Token Shuffle and Temporal Adaptive Position Encoding (TAPE), which are innovative approaches to compress visual tokens and enhance temporal awareness in video representations.

- **Quality**: The research demonstrates high quality through rigorous experimentation and ablation studies, showing significant improvements over existing benchmarks like Egoschema and VideoMME. The paper also presents a comprehensive dataset, TimePro, with 349k high-quality annotations.

- **Clarity**: The paper is well-structured and clearly articulated. Complex concepts such as Temporal Grounded Caption and TAPE are explained with clarity, making the paper accessible to readers. Figures and tables are effectively used to convey key results and comparisons.

- **Significance**: The proposed solutions for temporal grounding could have broad implications for video understanding tasks, and the zero-shot capabilities of VideoChat-T are particularly noteworthy, showing the potential to rival supervised expert models.

**Weaknesses:**

- **Generalization**: The paper primarily focuses on video datasets that are thematically similar. It is unclear how well TimeSuite generalizes to videos with significantly different content or from other domains, such as OpenEQA and CinePile. Expanding the dataset to include more diverse domains could strengthen the paper's claims.

- **Scalability and Efficiency**: The computational requirements for processing long videos are high. The paper could provide more insights into the scalability of TimeSuite and its efficiency, especially when dealing with very long videos or a large number of frames.

- **Longitudinal Performance**: The paper does not address how the model's performance degrades over time or with an increasing amount of video data. It would be beneficial to include studies on the long-term sustainability of the model's performance.

- **Qualitative Analysis**: While quantitative results are provided, a deeper qualitative analysis of the model's outputs, especially in cases of failure, could offer actionable insights into the model's reasoning process and areas for improvement.

**Questions:**

1. **Token Shuffle Mechanism**: Could the authors elaborate on the decision to use a token shuffling mechanism over other compression techniques, and provide a comparison of its performance against alternatives in terms of temporal consistency and computational efficiency?

2. **Temporal Adaptive Position Encoding (TAPE)**: It would be beneficial if the authors could discuss the potential limitations of TAPE in handling videos with highly variable or complex temporal dynamics, and whether there are any specific domains where TAPE excels or falls short.

3. **Data Diversity and Model Generalization**: How does the choice of datasets for TimePro impact the model's generalization capabilities? Are there any biases introduced by the current dataset composition that could limit the model's applicability to unseen video domains?

4. **Zero-Shot Performance vs. Fine-Tuning**: The paper mentions robust zero-shot capabilities. Could the authors provide insights into why there is a performance gap between zero-shot and fine-tuned models, and what aspects of the model or training process could be improved to bridge this gap?

5. **Long-Video Understanding Limitations**: Are there specific scenarios or types of long videos where VideoChat-T struggles? If so, could the authors suggest potential improvements or future work to address these limitations?

6. **Integration of Expert Model Capabilities**: The authors propose a future direction of integrating expert model capabilities into MLLMs. What are the authors' thoughts on the feasibility of this approach, and what challenges need to be overcome to achieve a unified generalist MLLM?

7. **Complex Output Formats**: For tasks requiring complex outputs, such as highlight detection, how can MLLMs be adapted to handle multiple discrete timestamps and saliency scores effectively? Are there plans to modify the model architecture or training process to better accommodate such tasks?

---

> ### Author Response · Authors · 2024-11-20
>
> We are very grateful for your constructive comments, which have been instrumental in improving our paper. The following is a point-by-point response to the reviewer's comments.
>
>
> **W1: Generalization: The paper primarily focuses on video datasets that are thematically similar. It is unclear how well TimeSuite generalizes to videos with significantly different content or from other domains, such as OpenEQA and CinePile. Expanding the dataset to include more diverse domains could strengthen the paper's claims.**
>
> Thank you for your suggestion. We are considering the inclusion of more diverse domain video data (such as OpenEQA and CinePile) in future updates to further strengthen our paper's claims. Our proposed TimeSuite employs fine-tuning data with a wide domain distribution and utilizes comprehensive evaluation benchmarks, demonstrating the robustness of our model to videos from different domains.
>
> **Fine-tuning Data:** Our proposed TimePro dataset comprises 9 different tasks, with video sources collected from 15 different datasets. Additionally, we adopted the Normal combination data for joint training, containing 4 task types with video sources from 5 different datasets. The fine-tuning dataset we used shows significant improvement in data quantity, data quality, and data diversity compared to previous temporal grounding fine-tuning datasets. This enhancement enables our approach to be robust to videos from various domains.
>
> **Evaluation Benchmarks:** After careful consideration, we chose MVBench, VideoMME, and Egoschema as our evaluation sets for general video understanding. These three datasets cover a wide range of video domains and diverse video length distributions. The performance metrics measured on these three benchmarks can well reflect the generalization of the model to videos from different domains.
>
> - MVBench is a comprehensive evaluation benchmark with 20 challenging video tasks, each containing 200 questions, covering a broad spectrum of temporal understanding skills from perception to cognition.
>
> - VideoMME features diverse video types, covering 6 major visual domains and 30 sub-domains to ensure broad scene generalization; in the temporal dimension, it includes short, medium, and long videos ranging from 11 seconds to 1 hour to achieve robust contextual dynamics.
>
> - EgoSchema, derived from Ego4D, primarily focuses on embodied scenes, containing 5000 first-person videos that cover a very broad range of natural human activities and behaviors.
>
> OpenEQA is indoor embodied data, mainly first-person videos, and its video domain distribution is quite similar to Egoschema. CinePile is oriented toward long movie videos, while VideoMME also includes evaluations targeting long movie videos. Our model has shown significant performance improvements on both Egoschema and VideoMME, which to some extent demonstrates the robustness of our proposed method to videos from these domains.

---

> ### Author Response · Authors · 2024-11-20
>
> **W2: Scalability and Efficiency: The computational requirements for processing long videos are high. The paper could provide more insights into the scalability of TimeSuite and its efficiency, especially when dealing with very long videos or a large number of frames.**
>
> Thank you for your valuable suggestion. We have supplemented our analysis with the scalability and computational efficiency of VideoChat-T.
>
> Due to the appropriate sequence compression achieved through Token Shuffle, the computational consumption of our VideoChat-T version is further reduced. By compressing 128 frames of video using Token Shuffle, the number of tokens input to the LLM is only 384, i.e., 3 tokens per frame, which significantly enhances the inference speed of the LLM. It shows a significant computational advantage compared to the latest well-known high-performance MLLMs like LLaVA-OneVision [1].
>
> As shown in the table below, under the same settings, the flops consumption of our VideoChat-T is 5.1% of LLaVA-OneVision, and the inference speed is only 0.63s, reaching real-time response levels. In terms of performance, VideoChat-T is also competitive. VideoChat-T significantly outperforms LLaVA-OneVision in temporal localization tasks, shows a slight advantage in MVBench, is roughly on par with LLaVA-OneVision in Egoschema, and only shows a significant disadvantage in VideoMME. Considering the significant savings in computational resources by VideoChat-T, we believe that the disadvantages in some datasets are acceptable. Furthermore, the faster inference speed facilitates the migration to practical applications such as online video inference.
>
> |  Method  | tokens (per  frame) | flops (128f) | Inference Time (128f) | Charades-STA | QVHighlight | MVBench | Egoschema | VideoMME |
> |--------|:----------------:|:-------------:|:----------------------:|:------------:|:-----------:|:-------:|:---------:|:--------:|
> |     -     |         -         |        -       |            -            |    IOU0.5    |     mAP     |   Avg   |    Full   |  w/o subs  |
> | LLaVA-OneVision [1] |        196       |   693.7T   |         4.95s         |      7.3     |    15.0    |   56.7  |    60.1   |   58.2   |
> |   VideoChat-T (Ours)   |         3        |     35.5T    |          0.63s         |     48.7     |     26.5    |   59.9  |    60.0   |   46.3   |
>
> For applications that need to handle ultra-long videos, since we only use 3 tokens per frame, our method theoretically requires only 30K context length to process a 10,000-frame video, saving considerable computational resources and achieving higher inference efficiency, making it possible to perform movie-level video inference on a single A100 GPU.
>
>
>
> **Citations:**
>
> [1] Li, Bo, et al. "Llava-onevision: Easy visual task transfer." arXiv preprint arXiv:2408.03326 (2024).

---

> ### Author Response · Authors · 2024-11-20
>
> **W3: Longitudinal Performance: The paper does not address how the model's performance degrades over time or with an increasing amount of video data. It would be beneficial to include studies on the long-term sustainability of the model's performance.**
>
>
> Thank you for your valuable suggestion. We have conducted an in-depth investigation into the performance degradation on MVBench and added the findings to Appendix F.2 (page 19) of the manuscript.
>
>
> | Method                    | post ft data           | data size | frame num | token num (per frame) | MVBench (AVG) |
> |---------------------------|------------------------|:---------:|:---------:|:---------------------:|:------------:|
> | VideoChat2                | -                      |     -     |     16     |           12          |     60.4     |
> | VideoChat2                | -                      |     -     |    128    |           12          |     42.1     |
> | VideoChat-T (Common_Init) | -                      |     -     |    128    |           3           |     25.3     |
> | VideoChat-T (Ours)        | -                      |     -     |    128    |           3           |     48.6     |
> | VideoChat-T (Ours)        | TimePro+Normal (Ours)  |   0.43M   |    128    |           3           |     59.9     |
> | VideoChat-T (Ours)        | TimePro+FullVideoChat2 |     2M    |    128    |           3           |     62.9     |
>
> **Our analysis revealed two primary factors contributing to the performance decline on MVBench:**
>
>
> - **Architectural Discrepancy:** The original VideoChat2 model was designed to process only 16 frames, leading to a mismatch in the learned feature distribution compared to the VideoChat-T architecture. As shown in the first two rows of the table, increasing the input frame number for VideoChat2 resulted in a significant performance drop. When initializing VideoChat-T with VideoChat2, performance was close to random due to the newly introduced randomly initialized layers. By applying efficient initialization to these new layers, we partially recovered the original capabilities of the model, bringing the MVBench performance of the un-trained VideoChat-T back to 48.6. After further fine-tuning, the short-video processing capability of VideoChat-T improved significantly, reaching 59.9.
>
> - **Fine-tuning Data Discrepancy:** We fine-tuned VideoChat-T using only 432K data, significantly less than the 2M non-grounded regular data used for training VideoChat2. The fine-tuning data for VideoChat2 primarily consisted of short videos of around ten seconds, which closely matched the length distribution of the MVBench evaluation videos, playing a crucial role in improving MVBench performance. To validate our hypothesis, we conducted additional experiments by training our VideoChat-T model using the TimePro and full VideoChat2 training data. We observed a improvement on the MVBench dataset.

---

> ### Author Response · Authors · 2024-11-20
>
> **W4: Qualitative Analysis: While quantitative results are provided, a deeper qualitative analysis of the model's outputs, especially in cases of failure, could offer actionable insights into the model's reasoning process and areas for improvement.**
>
> Thank you for your valuable suggestion. We have conducted comprehensive qualitative analyses to complement the quantitative results presented in this paper. Figure 4 (page 9) and Appendix H.1 (page 23) provide detailed qualitative analyses of our model's outputs on long-video QA, short-video QA, and captioning tasks.
>
> To further understand the model's limitations, we have performed an in-depth failure analysis, as presented in Appendix H.2 (page 24). Our results reveal that while the model achieves promising performance on many tasks, it still struggles with reasoning about complex logic, multi-scene long videos. The model's responses often lack sufficient detail to capture the intricate causal relationships within complex events. Due to the preponderance of single-turn, perceptual questions in our training data and the lack of multi-step reasoning data with complex logic, our model struggles to handle more challenging scenarios that demand intricate logical reasoning.
>
> > We also conducted a qualitative analysis of the shortcomings of VideoChat-T through examples. As shown in Figure 9, VideoChat-T performs poorly on examples with complex logic. In the left example, although VideoChat-T accurately identified the timing of the event, it failed to fully explain the motivation behind the man opening the isolation door, which was "to fight the hijackers of the space elevator, seize the controller, and thus save the people in the entire space elevator." In the right example, VideoChat-T correctly identified the event where Mr. Bean reached out to touch his desk mate's table, but it incorrectly explained the true reason for this action, which was "to cover up the fact that he was copying his desk mate's exam by pretending to wipe dust off the desk."
>
> These findings highlight the need for future research to explore more effective mechanisms for capturing long-range dependencies. To address this limitation, we propose constructing data in a chain-of-thought format to guide the model through multi-step reasoning, enabling it to delve deeper into the underlying motivations and causal relationships within a video.

---

> ### Author Response · Authors · 2024-11-20
>
> **Q1: Token Shuffle Mechanism: Could the authors elaborate on the decision to use a token shuffling mechanism over other compression techniques, and provide a comparison of its performance against alternatives in terms of temporal consistency and computational efficiency?**
>
>
>
> **Reason for Using Token Shuffle:**
>
> We chose the Token Shuffle mechanism over other methods for several reasons:
>
> - **Temporal Consistency:** Clustering methods struggle to maintain temporal consistency of video tokens, and pooling methods lead to performance loss. Token Shuffle preserves temporal information by connecting adjacent tokens in the channel dimension while compressing.
>
> - **Information Density:** Projecting visual encoding vectors from low to high dimensions does not necessarily increase information density. Token Shuffle enhances the amount of information carried by each token by stacking multiple tokens in the channel dimension.
>
> - **Model Generalization:** Introducing new compression methods may interfere with the original model. Token Shuffle avoids introducing additional randomly initialized parameters by inheriting the base model's projectors, thereby better retaining the original model's capabilities.
>
> - **Flexibility:** Pooling methods are less flexible. Token Shuffle can be further optimized through training, offering higher flexibility.
>
> **Temporal Consistency of Token Shuffle:**
>
> Our method mainly merges temporally adjacent tokens in the channel dimension during compression. The relative order of the long video sequence remains consistent before and after compression, ensuring temporal consistency.
>
> Clustering methods are unordered, potentially merging tokens with early and late temporal positions, thus disrupting the temporal distribution of the long video sequence and severely damaging temporal consistency.
>
> Clustering within a certain local temporal window can prevent the unordered merging of early and late tokens, potentially mitigating the damage to temporal consistency. For comparison, we replaced Token Shuffle with clustering within the local window defined by clip divisions in our ablation experiments. As shown in the table, even with the local window restriction, its performance is still inferior to our proposed Token Shuffle.
>
>
>
> | Model             | Egoschema | VideoMME | Charades-STA | QVHighlights |
> |-------------------|:---------:|:--------:|:------------:|:------------:|
> |                   |    Full   | w/o subs |  R@1 IOU=0.5 |     Hit@1    |
> | r/w pooling       |    59.8   |   44.8   |     40.3     |     47.3     |
> | r/w cluetering    |    59.5   |   45.0   |     39.8     |     40.1     |
> | VideoChat-T(Ours) |    60.0   |   46.3   |     48.7     |     54.1     |
>
>
> **Computational Efficiency of Different Methods:**
>
> At the same compression rate, the computational efficiency of different compression methods is essentially equivalent. This is because, for the same compression rate, fixed-length inputs are compressed into outputs of the same length, which are then fed into the LLM. The overall computational efficiency of the model mainly depends on the sequence length of the input to the LLM, and the compression layer itself constitutes only a very small portion of the model's total computational load. Therefore, our Token Shuffle method focuses more on the retention of the original sequence information and the temporal consistency of the sequence at the same compression rate, which will greatly impact the overall performance of the MLLM after sequence compression.

---

> ### Author Response · Authors · 2024-11-20
>
> **Q2: Temporal Adaptive Position Encoding (TAPE): It would be beneficial if the authors could discuss the potential limitations of TAPE in handling videos with highly variable or complex temporal dynamics, and whether there are any specific domains where TAPE excels or falls short.**
>
> Our designed TAPE is an external temporal encoding module that does not introduce significant additional computational load. It has the advantage of being plug-and-play, making it highly suitable for embedding relative positional encoding into one-dimensional temporal feature sequences, thereby enhancing the model's temporal perception capabilities without compromising its original abilities.
>
> A potential limitation of TAPE is its difficulty in processing ultra-long sequences in one go. This limitation might be addressed by processing in segments and then merging. Additionally, integrating the built-in RoPE method from LLMs (such as YaRN) and combining it with our proposed segmented TAPE usage could potentially aid in improving the model's performance.

---

> ### Author Response · Authors · 2024-11-20
>
> **Q3: Data Diversity and Model Generalization: How does the choice of datasets for TimePro impact the model's generalization capabilities? Are there any biases introduced by the current dataset composition that could limit the model's applicability to unseen video domains?**
>
> As mentioned in **W1**, in addition to utilizing TimePro with its 349K data points, we also integrated 82K of Normal data. TimePro encompasses a variety of tasks and a broad distribution of video sources, which to some extent ensures the generalization capability of the MLLM. Supplementing this with 82K high-quality Normal data, our proposed TimeSuite effectively retains the original capabilities of the base model, thereby preserving the generalization of MLLM. Additionally, our chosen evaluation benchmarks comprehensively cover videos from different domains, and the performance improvements on these benchmarks validate the robust generalization of our model.
>
> We have supplemented Appendix G.2 (page 22) with an exploration of whether temporal grounding data could impair the model's original generalization capabilities.
>
> |        FT Data        | MVBench (AVG) |
> |:---------------------:|:-------------:|
> |         TimeIT        |      54.7     |
> |     TimeIT+Normal     |      55.3     |
> |         Normal        |      56.1     |
> |        TimePro        |      57.4     |
> | TimePro+Normal (Ours) |      59.9     |
>
> Typically, the form of temporal grounding data is relatively singular, which can easily cause the model to overfit on temporal grounding tasks, leading to a loss of general question-answering capabilities and consequently damaging generalization. We compared the TimeIT dataset proposed in TimeChat [2] with our TimePro using MVBench. As shown in the table, fine-tuning using only TimeIT resulted in the poorest accuracy, and the performance of the model using TimeIT+Normal was also slightly lower than that using only Normal. This indicates that TimeIT does indeed impair the model's original performance (as shown in Figure 1 at the beginning of the paper, TimeChat lost some general question-answering capabilities after fine-tuning, and the right column shows that it outputs the localization time when given a general question).
>
> In contrast, our TimePro includes diverse data, covering 15 datasets spanning 9 different task types, which can mitigate the generalization loss caused by singular data to some extent. Moreover, our data combines grounding with various general tasks. For example, Grounded Captioning requires simultaneously outputting a detailed description of the corresponding video segment, and Reasoning Temporal Localization requires reasoning over the question at the same time. This greatly enhances the model's generalization, thereby reducing the damage to the model's original capabilities (i.e., short video accuracy). As shown in the table, the performance using only our TimePro is even higher than using only Normal, and the effect of using TimePro+Normal far surpasses all other combinations. This also demonstrates that our TimePro effectively avoids impairing the model's original performance.
>
>
> Citations:
>
> [2] Ren, Shuhuai, et al. "Timechat: A time-sensitive multimodal large language model for long video understanding." Proceedings of the IEEE/CVF Conference on Computer Vision and Pattern Recognition. 2024.

---

> ### Author Response · Authors · 2024-11-20
>
> **Q4: Zero-Shot Performance vs. Fine-Tuning: The paper mentions robust zero-shot capabilities. Could the authors provide insights into why there is a performance gap between zero-shot and fine-tuned models, and what aspects of the model or training process could be improved to bridge this gap?**
>
> The fine-tuned models utilize the training sets provided by the benchmarks during the training process. These benchmark training sets contain video distributions and task types that are the same as those in the test set. This same-domain training data enables the model to perform better on the test set but may result in a loss of generalization to other tasks. It is very common and reasonable for zero-shot performance to be lower than fine-tuned performance in most fields. To bridge the performance gap between zero-shot and fine-tuned models, we can enhance the model's generalization by incorporating broader and more comprehensive training data and employing more reasonable training methods.

---

> ### Author Response · Authors · 2024-11-20
>
> **Q5: Long-Video Understanding Limitations: Are there specific scenarios or types of long videos where VideoChat-T struggles? If so, could the authors suggest potential improvements or future work to address these limitations?**
>
> As mentioned in **W4**, our method shows insufficiency in handling videos with complex logic, making it difficult to reason about events with multi-step complex logic in long videos. One potential improvement could be to introduce a chain-of-thought approach to guide the model in step-by-step reasoning, thereby enabling it to correctly answer long video questions with complex logic.

---

> ### Author Response · Authors · 2024-11-20
>
> **Q6: Integration of Expert Model Capabilities: The authors propose a future direction of integrating expert model capabilities into MLLMs. What are the authors' thoughts on the feasibility of this approach, and what challenges need to be overcome to achieve a unified generalist MLLM?**
>
> Thank you for your interest in our insights. In Appendix G.1 (page 21) of the manuscript, we discuss in detail the feasibility of enhancing the comprehensive video understanding capabilities of MLLMs by integrating more expert tasks. By analyzing the limitations of MLLMs in handling expert tasks, we specifically elucidate the challenges of incorporating expert tasks into MLLMs and propose potential solutions to address these challenges.
>
> We believe that MLLMs have strong zero-shot capabilities and, with appropriate fine-tuning, can rival the current state-of-the-art expert models. However, there are challenges in directly integrating the capabilities of expert models into MLLMs, including issues with complex output formats and the bottleneck of language representations, as well as the need for diverse data.
>
> Currently, an effective and relatively easy-to-implement solution is to introduce expert models as external decoders. Nevertheless, we aim to internalize the expert tasks as inherent capabilities of MLLMs without relying on external decoders. This primarily requires focusing on **two key aspects**:
>
> - **Simplifying Complex Output Formats:** Researching how to convert the complex output formats of expert tasks into language representations that LLMs can understand, or designing special processing procedures to simplify these tasks.
>
> - **Increasing Data Diversity:** Designing diverse data formats to enrich the training data of MLLMs, thereby enhancing their generalization capabilities across different tasks.

---

> ### Author Response · Authors · 2024-11-20
>
> **Q7: Complex Output Formats: For tasks requiring complex outputs, such as highlight detection, how can MLLMs be adapted to handle multiple discrete timestamps and saliency scores effectively? Are there plans to modify the model architecture or training process to better accommodate such tasks?**
>
> Currently, we guide the MLLM to follow a specified format by adding prompt words with standard response formats to the MLLM's input. The prompt we use is as follows:
>
> > Please find the highlight contents in the video described by a sentence query, determining the highlight timestamps and its saliency score on a scale from 1 to 5. The output format should be like: 'The highlight timestamps are in the 52, 54, 56, 58 seconds. Their saliency scores are 3.0, 3.0, 3.0, 3.0'. Now I will give you the sentence query: {}. Please return the query-based highlight timestamps and salient scores.
>
>
> This prompt guides the model to output the highlight timestamps and corresponding saliency scores as specified. However, if there are too many highlight moments in the video (e.g., more than 20), the accuracy of the output significantly decreases, which is precisely what we referred to in the paper as the language representation bottleneck.
>
> An obvious solution is to use an external expert model decoder for the MLLM, for instance, by sending the feature tokens output by the MLLM to an expert detection head. Additionally, there are feasible methods that do not rely on an expert decoder. For example, we might refer to Hawkeye's iterative output method or segment long videos for processing, ensuring that only a portion of the highlight timestamps and corresponding saliency scores are output in each iteration.
>
>
> I would like to express my sincere gratitude for your thoughtful comments and suggestions. Your feedback has been invaluable in improving this manuscript.

---

> > ### Comment · Reviewer_epk2 · 2024-11-22
> >
> > Thanks for your detailed and well-reasoned responses. I have carefully reviewed the concerns raised by other reviewers, the Public Comment, and the associated discussions and details. I find no major flaws in this paper. The innovation is commendable, and the research problem involves a topic of significant interest to the community. The quality of the work is solid and comprehensive.
> >
> > Given these considerations, I have decided to raise my score from 6 to 8. This paper provides valuable insights for advancing research in long video understanding. Please ensure that all discussed points are incorporated into the final version of the paper, and the code is made open-source.

---

### Official Review · Reviewer_x14u · 2024-11-01

**Soundness:** 3
**Presentation:** 2
**Contribution:** 2
**Rating:** 6
**Confidence:** 5

**Summary:**

This paper presents TimeSuite, a novel approach designed to enhance the understanding of long-form videos using existing short-form MLLMs. TimeSuite includes an efficient framework for processing long video sequences, a high-quality dataset for grounded tuning, and a new instruction tuning task to integrate grounding supervision in a traditional QA format. Experimental results demonstrate the advantage of  the proposed method.

**Strengths:**

The proposed method is both systematic and effective. The experiments are comprehensive and detailed, and the results of the method are convincing.

**Weaknesses:**

Although the proposed method includes specific designs tailored to the problem, the novelty is limited. It is a relatively engineering-focused paper. The systematic description is quite detailed, but it lacks a deeper discussion of the underlying ideas, theories and principles. Additionally, due to the design of auxiliary tasks and datasets, the comparison with baseline methods may be somewhat unfair.

**Questions:**

It would be beneficial to include a more in-depth discussion of theories and principles in the manuscript.

---

> ### Author Response · Authors · 2024-11-19
>
> We would like to express our sincere gratitude for your review. Your suggestions will greatly help us improve our paper.
>
>
> **W1: Although the proposed method includes specific designs tailored to the problem, the novelty is limited. It is a relatively engineering-focused paper. The systematic description is quite detailed, but it lacks a deeper discussion of the underlying ideas, theories and principles.**
>
> **R1:**
>
> We appreciate the reviewer's careful assessment of our work. In the revised manuscript, we will further highlight the novelty of our approach and delve deeper into the underlying theories and principles.
>
>
> We believe the novelty of TimeSuite can be summarized in three key aspects:
>
> **1. Architectural Novelty:** Our proposed Token Shuffle enables efficient compression of long video sequences without sacrificing significant detail, while TAPE enhances the model's temporal awareness. The combination of Token Shuffle and TAPE achieves an extreme compression ratio of only 3 tokens per frame. As shown in the table below, our method is computationally efficient, requiring only 35.5T flops for inferring 128 frames and achieving an inference speed of 0.63s, making it suitable for real-time video inference tasks.
>
> |  Method  | tokens (per  frame) | flops (128f) | Inference Time (128f) | Charades-STA | QVHighlight | MVBench | Egoschema | VideoMME |
> |--------|:----------------:|:-------------:|:----------------------:|:------------:|:-----------:|:-------:|:---------:|:--------:|
> |     -     |         -         |        -       |            -            |    IOU0.5    |     mAP     |   Avg   |    Full   |  w/o subs  |
> | LLaVA-OneVision [1] |        196       |   693.7T   |         4.95s         |      7.3     |    15.0    |   56.7  |    60.1   |   58.2   |
> |   VideoChat-T (Ours)   |         3        |     35.5T    |          0.63s         |     48.7     |     26.5    |   59.9  |    60.0   |   46.3   |
>
>
> **2. Data Novelty:** To address the lack of high-quality data for MLLMs, we introduce TimePro, a more diverse and comprehensive dataset. Additionally, we propose Temporal Grounded Caption to mitigate hallucinations in MLLMs.
>
>
> **3. Conceptual Novelty: Most importantly, the design of our architecture and dataset is driven by our core contribution: providing a new perspective for MLLM research.**
>   - **We demonstrate that MLLMs, without relying on external decoders, can achieve performance comparable to expert models on specific tasks while maintaining strong generalization abilities and zero-shot capabilities.**
>   - **By incorporating expert tasks, we enhance the MLLM's comprehensive long-video understanding capabilities. Our experiments validate the feasibility of improving MLLMs' overall performance by integrating expert tasks.**
>
> **Underlying Ideas:** Our goal is to maintain the model's general QA capabilities while introducing time-grounded expert tasks to reduce hallucinations and improve the model's overall performance in long-video understanding. We elaborate on the motivation behind our work in the introduction of the paper.
>
> **Theories and Principles:** We maximize the reuse of pre-trained MLLM parameters and design a highly diverse dataset to prevent a loss of the model's original general-purpose performance. Building upon this foundation, we design a compression module specifically for long videos and fine-tune it with grounded data. This validates the feasibility of improving MLLM's comprehensive performance by incorporating expert model capabilities.
>
> We have included a more in-depth discussion of these points in the appendix, specifically in the Extra Ablation and Discussion sections (please see **Q1** for details).
>
>
> **Citations:**
>
> [1] Li, Bo, et al. "Llava-onevision: Easy visual task transfer." arXiv preprint arXiv:2408.03326 (2024).

---

> ### Author Response · Authors · 2024-11-19
>
> **W2: Additionally, due to the design of auxiliary tasks and datasets, the comparison with baseline methods may be somewhat unfair.**
>
> **R2:**
>
> To address the reviewer's concern regarding potential bias due to the design of auxiliary tasks and datasets, we conducted an additional experiment. We fine-tuned TimeChat using the exact same fine-tuning data and auxiliary tasks as VideoChat-T. The results, presented in table below, demonstrate that while TimeChat shows improvements across all evaluation benchmarks, its performance still falls short of VideoChat-T. This suggests that both the fine-tuning architecture and the quality and diversity of the training data play crucial roles in model performance.
>
> |    Method   	|      Data      	| Charades-STA 	| QVHighlight 	| MVBench 	| Egoschema 	| VideoMME 	|
> |-----------|--------------|:------------:|:-----------:|:-------:|:---------:|:--------:|
> |             	|                	|    IOU0.5    	|     mAP     	|   Avg   	|    Full   	| w/o subs 	|
> |   TimeChat  	|  TimeIT+Valley 	|     32.2     	|     14.5    	|   38.5  	|    33.0   	|   30.2   	|
> |   TimeChat  	| TimePro+Normal 	|     34.2     	|     16.3    	|   41.6  	|    38.9   	|   33.4   	|
> | VideoChat-T 	| TimePro+Normal 	|     48.7     	|     26.5    	|   59.9  	|    60.0   	|   46.3   	|
>
> To further substantiate our claims, we conducted comprehensive ablation studies, detailed in Tables 3, 4, and 5 in the paper (page 9 & 10), which isolate the contributions of our proposed architecture and dataset. These results provide a more fair comparison and highlight the significance of each component in achieving our results.

---

> ### Author Response · Authors · 2024-11-19
>
> **Q1: It would be beneficial to include a more in-depth discussion of theories and principles in the manuscript.**
>
> **A1:**
>
> Building upon the underlying ideas and basic theories and principles discussed in **W1**, we have included additional in-depth discussions on these theoretical underpinnings in the revised manuscript.
>
> **1. Preserving the model's original generalization ability:**
>   - Appendix F.2 (page 19) provides a more in-depth analysis of the performance degradation of the model on short videos, accompanied by additional ablation studies to investigate the two key factors contributing to the loss of the model's original capabilities.
>     >Based on the above, we can conclude the fundamental reasons affecting the model's foundational generalization capabilities. When a model undergoes adjustments, the learned original distribution may not perfectly match the new architecture, making the efficient initialization of new layers crucial. The features learned from the original dataset might be forgotten due to changes in various parameters. Utilizing a more comprehensive and diverse dataset for fine-tuning can restore and even further enhance performance.
>
>   - Appendix G.2 (page 22) delves into the reasons why previous expert task data can lead to a loss of generalization ability in MLLMs, and how we can mitigate the negative impact of novel expert task data on the model's original capabilities through data design.
>     >Overall, using a single type of expert task training data can easily lead to model overfitting, resulting in significant loss of the model's original capabilities. To preserve the model's foundational generalization abilities, it is essential to use diversified training data. Additionally, incorporating data of various types and distributions, such as text, images, and videos, can further enhance the model's generalization capabilities.
>
> **2. Enhancing the model's comprehensive capabilities through the introduction of expert tasks:**
>   - Appendix G.1 (page 21) provides an in-depth exploration of the feasibility of enhancing the MLLM's comprehensive video understanding capabilities by incorporating more expert tasks. By analyzing the limitations of MLLMs in handling expert tasks, we specifically discuss the challenges of integrating expert tasks into MLLMs and outline potential solutions to address these challenges.
>     > Through the integration of diverse expert tasks and the optimization of language representations, MLLMs can achieve substantial improvements in their overall capabilities. This allows them to effectively comprehend and address complex tasks, rivaling or even exceeding the performance of specialized expert models within specific domains. Looking ahead, MLLMs have the potential to evolve into highly versatile AI models, transcending traditional conversational and QA capabilities. They will be equipped to handle a wide range of complex expert tasks across various domains, such as vision, language, and reasoning.
>
> Additionally, the computational efficiency analysis in Appendix C (page 17) confirms the phenomenon that the model can maintain relatively robust performance even under extremely high compression ratios, and we have conducted a related discussion.
> > Moreover, our model's ability to maintain reasonable performance under high compression ratios suggests that the token embedding spaces of contemporary models may be characterized by considerable feature redundancy. This observation presents a promising avenue for future research, as efficient techniques for compressing or discarding redundant features could substantially reduce computational costs without sacrificing model performance, enabling longer context reasoning.
>
>
> We would like to express our sincere gratitude for your valuable feedback, which has been instrumental in our research. In subsequent revisions of the manuscript, we will focus on highlighting the novelty of our work and continue to enhance the depth of our discussions on the underlying theories and principles.

---

### Official Review · Reviewer_M5C1 · 2024-11-04

**Soundness:** 2
**Presentation:** 2
**Contribution:** 3
**Rating:** 3
**Confidence:** 5

**Summary:**

The paper introduces TimeSuite, a framework designed to enhance Multimodal Large Language Models (MLLMs) for long video understanding. It proposes a new long-video MLLM called VideoChat-T, featuring token shuffling and Temporal Adaptive Position Encoding (TAPE) for better temporal awareness. TimeSuite includes a specialized instruction tuning dataset, TimePro, comprising 349,000 annotations across nine tasks, with a focus on Temporal Grounded Captioning for accurate video descriptions with timestamps. Experiments show that TimeSuite improves long video comprehension, achieving notable performance gains on benchmarks and demonstrating strong zero-shot grounding capabilities, rivaling traditional supervised models after fine-tuning.

**Strengths:**

- This paper propose TimeSuite, a collection of new designs to improve the long video understanding capability of the existing short-form MLLMs

**Weaknesses:**

- The paper does not adequately compare its results against notable existing works, such as Qwen2-VL and Llava-OneVision. A comprehensive comparison would help contextualize the contributions and highlight any advantages or shortcomings.
- The dataset used in the experiments, TimePro is highly imbalanced (Figure 3). It is unclear whether the authors have explored different data ratios or configurations. Investigating this aspect could provide insights into the robustness of the proposed method and its adaptability to varying data distributions.

**Questions:**

See weaknesses

---

> ### Author Response · Authors · 2024-11-24
>
> We are very grateful for your time and expertise in reviewing our submission.
>
> **W1: The paper does not adequately compare its results against notable existing works, such as Qwen2-VL and Llava-OneVision. A comprehensive comparison would help contextualize the contributions and highlight any advantages or shortcomings.**
>
> Thank you for your insightful comments. We compare our method with the aforementioned methods, and also employ TimeSuite to Llava-OneVision to further validate its effectiveness and generalization.
>
>
> **Comparison with the latest prominent methods:**
>
> | Method             | Tokens      | flops         | Inference Time               | Charades-STA | QVHighlight | MVBench | Egoschema | VideoMME |
> |--------------------|-------------|---------------|------------------------------|:------------:|:-----------:|:-------:|:---------:|:--------:|
> |                    | (per frame) | (128 frames) |  (128f & on single A100 GPU) |    IOU0.5    |     mAP     |   Avg   |    Full   | w/o subs |
> | Qwen2-VL            | 138         | 929.8T        | Out Of Memory                |     15.0     |     13.0    |   67.0  |    66.7   |   63.3   |
> | Llava-OneVision    | 196         | 693.7T        | 4.95s                        |      7.3     |    15.0    |   56.7  |    60.1   |   58.2   |
> | VideoChat-T (Ours) | 3           | 35.5T         | 0.63s                        |     48.7     |     26.5    |   59.9  |    60.0   |   46.3   |
>
> Recently released MLLMs (e.g., Qwen2-VL, Llava-OneVision) have been trained on larger datasets and with higher computational resources, leading to strong foundational QA capabilities but weaker temporal grounding abilities. Our proposed method can enhance the grounding capabilities of MLLMs at a lower cost. Worth emphasizing is that our VideoChat-T further reduces computational costs through the application of Token Shuffle for appropriate sequence compression. Compared to methods like Llava-OneVision and Qwen2-VL, our model offers a significant computational advantage. As shown in the table, under the same settings, **only 3 tokens per freame lead to the flops of our VideoChat-T are 5.1% of Llava-OneVision, and the inference speed is only 0.63 seconds, achieving real-time response.** In terms of performance, VideoChat-T is also competitive. VideoChat-T significantly outperforms Llava-OneVision on temporal grounding tasks, has a slight advantage on MVBench, and performs similarly to Llava-OneVision on Egoschema, showing a disadvantage only on VideoMME. Considering the significant computational resource savings of VideoChat-T, we believe the disadvantages on some datasets are acceptable. Additionally, the faster inference speed facilitates migration to practical applications such as online video understanding.
>
> We hope to enhance the long-video understanding capabilities of MLLMs through temporal grounded fine-tuning. So we initially employed a lightweight baseline, as the training process of VideoChat2 is relatively computationally efficient, enabling us to complete fine-tuning more rapidly and conduct more comprehensive ablation studies. Given the significant gap in foundational QA performance between our baseline (VideoChat2) and these computationally intensive MLLMs (e.g., Qwen2-VL, Llava-OneVision), we did not directly compare VideoChat-T with the latest high-computation models in the initial manuscript. Instead, we validated our claims by observing performance improvements of our method (VideoChat-T) over the baseline approach (VideoChat2), as shown in Table 1&2 in the manuscript (page 7&8).
>
> **Further validation of the effectiveness and generalization of TimeSuite:**
>
> | Method                       | Charades-STA | QVHighlight | VideoMME | MLVU | MVBench |
> |------------------------------|:------------:|:-----------:|:--------:|:----:|:-------:|
> |                              |    IOU0.5    |     mAP     | w/o subs |  Avg |   Avg   |
> | Llava-OneVision (baseline) |      7.3     |     15.0    |   58.2   | 64.7 |   56.7  |
> | Llava-OneVision-T (Ours)   |       42.5       |       21.7      |   61.4   | 69.4 |   56.1  |
>
> To verify the robustness of our TimeSuite for other MLLMs, we transferred our method to Llava-OneVision. The table shows the performance changes of Llava-OneVision after applying our TimeSuite. It can be seen that when we apply the full set of methods in TimeSuite to Llava-OneVision, the model's performance on two different long-video evaluation benchmarks improves (+3.2 on VideoMME and +4.7 on MLVU), effectively demonstrating the robustness of our TimeSuite for different MLLMs.

---

> ### Author Response · Authors · 2024-11-24
>
> **W2: The dataset used in the experiments, TimePro is highly imbalanced (Figure 3). It is unclear whether the authors have explored different data ratios or configurations. Investigating this aspect could provide insights into the robustness of the proposed method and its adaptability to varying data distributions.**
>
> Thank you for your suggestions. We give initial explorations of data configrations of TimePro as follows.
>
> | Method                       | MVBench | Egoschema | VideoMME | Charades-STA | QVHighlight |
> |------------------------------|:-------:|:---------:|:--------:|:------------:|:-----------:|
> |                              |   Avg   |    Full   | w/o subs |    IOU=0.5   |     mAP     |
> | TimePro615K+Normal82K (old version)       |   60.0  |    61.0   |   46.3   |     45.4     |     25.7    |
> | TimePro349K+Normal82K (Ours) |   59.9  |    60.0   |   46.3   |     48.7     |     26.5    |
>
> In the early versions of TimePro, we employed datasets comprising 309K Multi-format Temporal Grounding instances, 150K Temporal Grounded Caption instances and other data. Through extensive experimentation, we discovered that removing low-quality data while retaining high-quality instances could significantly reduce training time without compromising performance. Consequently, we pruned these two part datasets to 100K and 93K instances, respectively. The data distribution presented in the paper represents the optimized and relatively balanced configuration we arrived at.
>
> | Normal | TimeIT | TGC | HD | MTG | RTL | Egoschema | VideoMME | Charades-STA | QVHighlights |
> |:------:|:------:|:---:|:--:|:---:|:---:|:---------:|:--------:|:------------:|:------------:|
> |        |        |     |    |     |     |    Full   | w/o subs |  R@1 IOU=0.5 |     Hit@1    |
> |    √   |        |     |    |     |     |    56.6   |   42.6   |      8.0     |     24.4     |
> |    √   |    √   |     |    |     |     |    57.8   |   43.6   |     32.2     |     25.2     |
> |    √   |    √   |  √  |    |     |     |    58.3   |   44.0   |     39.1     |     33.9     |
> |    √   |    √   |  √  |  √ |     |     |    59.8   |   44.9   |     41.9     |     43.8     |
> |    √   |    √   |  √  |  √ |  √  |     |    60.0   |   45.1   |     45.8     |     48.3     |
> |    √   |    √   |  √  |  √ |  √  |  √  |    60.0   |   46.3   |     48.7     |     54.1     |
>
> We verify the effectiveness of TimePro in the table above (also given in Table 5 of Sec. 4.5, page 10 in the manuscript). It demonstrates that as we incrementally add new data, the model's performance on temporal grounding and long videos consistently improves. The effectiveness of a composite dataset in enhancing model capabilities is influenced by various factors, including data quantity, quality, and task types. An imbalanced data volume distribution does not necessarily imply a low-quality composite dataset. The inclusion of a smaller quantity of high-quality samples can notably improve the overall model performance. To fully clarify data ratios or other factors in TimePro asks huge computes. We are working on it continuously and will include more analysis in the final version.
>
>
>
>
>
> We would like to thank you again for your time and valuable feedback. Your comments have greatly enhanced the quality of our work.

---

> ### Author Response · Authors · 2024-12-02
> **Gentle Reminder of Our Response**
>
> Thank you once again for your insightful feedback and for contributing to the improvement of our work! We would like to gently remind you that we have responded to the concerns you previously raised. As the discussion period nears its end, we would be grateful if you could share any remaining questions or issues that we could address before the deadline.

---

### Official Review · Reviewer_N2GP · 2024-11-04

**Soundness:** 3
**Presentation:** 3
**Contribution:** 3
**Rating:** 6
**Confidence:** 4

**Summary:**

This paper introduces TimeSuite, a collection of designs focusing on efficient architecture, high-quality data, and a novel instruction-tuning task. Building on TimeSuite, the authors propose a long-video multimodal large language model (MLLM) named VideoChat-T, which demonstrates robust zero-shot temporal grounding capabilities and significantly outperforms existing state-of-the-art MLLMs. After fine-tuning, it performs comparably to traditional supervised expert models.

**Strengths:**

1. The paper is well-written. The proposed method is well-illustrated and easy to follow.
2. The contributions include a strong video LLM (VideoChat-T) and a temporal-centric instruction-tuning dataset (TimePro), both of which are substantial advancements in the field.
3. The authors conduct a thorough evaluation and ablation study to validate the effectiveness of the proposed model.

**Weaknesses:**

While I do not see major flaws in this paper, I note that the technical novelty is relatively limited.

There are two main modules in VideoChat-T: (1) the VL-connector with token shuffling, and (2) temporal adaptive position encoding (TAPE).
However, the VL-connector's approach of concatenating tokens in the channel dimension and then compressing the channel dimension with a linear layer has been previously utilized in Qwen2-vl (I am unsure who proposed this operation first; please correct me if you know). For TAPE, the design of the position embedding is derived from CPVT. These factors somewhat limit the technical novelty of this work.

**Questions:**

1. Why is the operation of “concatenating tokens in the channel dimension and then compressing the channel dimension with a linear layer” referred to as “token shuffle”?
2. Can you use the same training data (e.g., TimePro) to train other VideoLLMs (e.g., TimeChat) and then compare their performance? This would help eliminate the impact of training data, allowing for a clearer assessment of how performance is associated with model design.

---

> ### Author Response · Authors · 2024-11-18
>
> (Due to the length of the response exceeding the limit, we reply in two parts. This is the first part.)
>
> Thanks for your kind review. We provide our feedbacks as follows.
>
> **W1：
> While I do not see major flaws in this paper, I note that the technical novelty is relatively limited.
> There are two main modules in VideoChat-T: (1) the VL-connector with token shuffling, and (2) temporal adaptive position encoding (TAPE). However, the VL-connector's approach of concatenating tokens in the channel dimension and then compressing the channel dimension with a linear layer has been previously utilized in Qwen2-vl (I am unsure who proposed this operation first; please correct me if you know). For TAPE, the design of the position embedding is derived from CPVT. These factors somewhat limit the technical novelty of this work.**
>
> **R1：**
>
> Thank you for your constructive comments. We believe the novelty of TimeSuite lies in several key aspects:
>
> **1. Structural Novelty:** While our structural design was inspired by previous work, it includes critical differences:
>   - The Token Shuffle was inspired by pixel shuffle in the field of super-resolution [1], [2]. Unlike other works, our Token Shuffle is highly transferable, efficiently initialized with the original model’s parameters, equivalent to AvgPooling at the start of the training, and gradually optimized through fine-tuning, minimizing damage to the original feature distribution.
>   - TAPE was inspired by CPVT, which is used for two-dimensional adaptive image spatial position encoding in the image backbone. TAPE extends CPVT to specifically enhance one-dimensional temporal perception in MLLM, proving its feasibility in temporal modeling within MLLM.
>   - Combining Token Shuffle and TAPE achieves an extreme compression ratio (only 3 tokens per frame). As shown in the table below, our method is computationally efficient, with flops of only 35.5T for inferring 128 frames, and an inference speed of just 0.63s, enabling real-time responsiveness.
>
> |  Method  | tokens (per  frame) | flops (128f) | Inference Time (128f) | Charades-STA | QVHighlight | MVBench | Egoschema | VideoMME |
> |--------|:----------------:|:-------------:|:----------------------:|:------------:|:-----------:|:-------:|:---------:|:--------:|
> |     -     |         -         |        -       |            -            |    IOU0.5    |     mAP     |   Avg   |    Full   |  w/o subs  |
> | LLaVA-OneVision [3] |        196       |   693.7T   |         4.95s         |      7.3     |    15.0    |   56.7  |    60.1   |   58.2   |
> |   VideoChat-T (Ours)   |         3        |     35.5T    |          0.63s         |     48.7     |     26.5    |   59.9  |    60.0   |   46.3   |
>
> **2. Data Novelty:** To address the lack of high-quality grounding data for MLLM, we proposed a more diverse dataset and data construction method:
>   - We introduced the comprehensive dataset TimePro, which includes 9 task types with video sources from 15 different datasets.
>   - We designed a novel Temporal Grounded Caption fine-tuning task to effectively mitigate hallucinations in MLLM.
>
> **3. Valuable Findings (the most crucial novelty of this paper):** Structural and data designs aim to better validate our propositions. The core contribution of TimeSuite is offering a new perspective for MLLM research:
>   - **Without relying on any external expert decoder, MLLM can achieve expert-level performance in specific tasks while maintaining considerable generalization QA capability and strong zero-shot capabilities.**
>   - **The introduction of expert tasks enhances the comprehensive understanding of long videos by MLLM. We validated the feasibility of enhancing MLLM’s comprehensive capabilities by integrating expert tasks.**
>
> We will highlight the innovations of this paper in subsequent updates to the manuscript.

---

> ### Author Response · Authors · 2024-11-18
>
> (Due to the length of the response exceeding the limit, we will reply in two parts. This is the second part.)
>
>
>
> **Q1:
> Why is the operation of “concatenating tokens in the channel dimension and then compressing the channel dimension with a linear layer” referred to as “token shuffle”?**
>
>
> **A1:**
> Our method is inspired by Pixel Shuffle [1], [2], which is an upsampling technique in image super-resolution that reorganizes features along the channel dimension. Our approach merges and reorganizes the tokens compressed by QFormer along the channel dimension, compressing the entire long token sequence into a shorter token sequence. Therefore, we named it Token Shuffle.
>
>
> **Q2:
> Can you use the same training data (e.g., TimePro) to train other VideoLLMs (e.g., TimeChat) and then compare their performance? This would help eliminate the impact of training data, allowing for a clearer assessment of how performance is associated with model design.**
>
> **A2:**
> Thank you for your suggestion. We have supplemented the performance on five benchmarks of TimeChat fine-tuned with our training data. It can be observed that TimeChat fine-tuned with our training data shows improvements across all benchmarks. However, its performance still lags significantly behind VideoChat-T. This indicates that an efficient fine-tuning architecture design and high-quality, diverse datasets are both essential and complementary.
>
> |    Method   	|      Data      	| Charades-STA 	| QVHighlight 	| MVBench 	| Egoschema 	| VideoMME 	|
> |-----------|--------------|:------------:|:-----------:|:-------:|:---------:|:--------:|
> |             	|                	|    IOU0.5    	|     mAP     	|   Avg   	|    Full   	| w/o subs 	|
> |   TimeChat  	|  TimeIT+Valley 	|     32.2     	|     14.5    	|   38.5  	|    33.0   	|   30.2   	|
> |   TimeChat  	| TimePro+Normal 	|     34.2     	|     16.3    	|   41.6  	|    38.9   	|   33.4   	|
> | VideoChat-T 	| TimePro+Normal 	|     48.7     	|     26.5    	|   59.9  	|    60.0   	|   46.3   	|
>
> Thank you again for your feedback, which is valuable in helping us improve the quality of our manuscript.
>
>
> **Citations:**
>
> [1] Shi, Wenzhe, et al. "Real-time single image and video super-resolution using an efficient sub-pixel convolutional neural network." Proceedings of the IEEE conference on computer vision and pattern recognition. 2016.
>
> [2] Pytorch. "Docs>torch.nn>PixelShuffle." © Copyright 2023, PyTorch Contributors. https://pytorch.org/docs/stable/generated/torch.nn.PixelShuffle.html
>
> [3] Li, Bo, et al. "Llava-onevision: Easy visual task transfer." arXiv preprint arXiv:2408.03326 (2024).

---

> ### Comment · Reviewer_N2GP · 2024-11-22
> **Official Comment by Reviewer N2GP**
>
> Thank you for your detailed responses. My main concerns have been addressed. Although the technical novelty may not be particularly strong, the work is solid. I would like to increase my score to a 7, if such an option were available in the rating choices. Overall, I maintain a positive evaluation for this paper.

---

### Official Review · Reviewer_61dM · 2024-11-10

**Soundness:** 3
**Presentation:** 3
**Contribution:** 3
**Rating:** 6
**Confidence:** 4

**Summary:**

The paper introduces TimeSuite, a set of designs to adapt short-form video multimodal large language models (MLLMs) for long video understanding. It includes a new framework for processing long video sequences, a high-quality dataset (TimePro) for grounded tuning, and an instruction tuning task to incorporate grounding supervision. The proposed model, VideoChat-T, uses token shuffling for compression and Temporal Adaptive Position Encoding (TAPE) for enhanced temporal awareness. A new task type, Temporal Grounded Caption, is introduced to improve video descriptions and timestamp prediction. Experimental results show significant improvements on benchmarks and robust zero-shot temporal grounding capabilities.

**Strengths:**

1. This work is highly complete. The contributions are multiple folds: VideoChat-T framework, TimePro dataset, Temporal Grounded Caption task, and extensive experiments and analysis.

2. Significant improvements on long video processing shown in Table 2.

3. Interesting TAPE, utilizing zero-padding anchors and gradual transmission of relative temporal positional encoding.

**Weaknesses:**

1. The VideoChat2 baseline shows a weak performance with 39.5 accuracy in VideoMME. It would be more convincing to use a stronger 7B model for evaluation, such as onevision-7B, which achieves 58.2 accuracy.

2. The short-term performance on MVbench drops require a deeper investigation rather than mere explanations.

3. Instead of using implicit TAPE, consider implementing a more explicit solution like 3D RoPE (e.g., Qwen2-VL) and conduct an ablation study.

**Questions:**

Could training the model on both temporal and non-temporal grounding data mitigate performance loss in short-term videos? Why does temporal grounding data lead to accuracy loss in short-term videos? Despite this, why is short-term accuracy on VideoMME still improved?

---

> ### Author Response · Authors · 2024-11-21
>
> We would like to extend our heartfelt gratitude to the reviewer for your insightful comments and valuable feedback. Your questions have significantly contributed to the improvement of our work, and we appreciate the opportunity to address them.
>
> **W1: The VideoChat2 baseline shows weak performance with 39.5 accuracy in VideoMME. It would be more convincing to use a stronger 7B model for evaluation, such as Onevision-7B, which achieves 58.2 accuracy.**
>
> We thank the reviewer for the insightful comment. Due to constraints on GPU resources, we initially selected the computationally efficient VideoChat2 as our baseline model. This choice enabled rapid fine-tuning and comprehensive ablation studies. We are also supplemented experiments on LLaVA-OneVision [1], a more computationally expensive model.
>
>
> **Comparison of computational efficiency and performance between VideoChat-T and LLaVA-OneVision:**
>
> |  Method  | tokens (per  frame) | flops (128f) | Inference Time (128f) | Charades-STA | QVHighlight | MVBench | Egoschema | VideoMME |
> |--------|:----------------:|:-------------:|:----------------------:|:------------:|:-----------:|:-------:|:---------:|:--------:|
> |     -     |         -         |        -       |            -            |    IOU0.5    |     mAP     |   Avg   |    Full   |  w/o subs  |
> | LLaVA-OneVision [1] |        196       |   693.7T   |         4.95s         |      7.3     |    15.0    |   56.7  |    60.1   |   58.2   |
> |   VideoChat-T (Ours)   |         3        |     35.5T    |          0.63s         |     48.7     |     26.5    |   59.9  |    60.0   |   46.3   |
>
> While not explicitly highlighted in the original manuscript, it is noteworthy that the application of Token Shuffle has enabled us to substantially reduce the computational requirements of VideoChat-T, thereby providing it with a considerable advantage over LLaVA-OneVision. Under equivalent experimental conditions, VideoChat-T consumes merely **3 tokens per frame**, corresponding to a mere **5.1% of the flops** incurred by LLaVA-OneVision. Additionally, it achieves an inference speed of only 0.63 seconds, demonstrating real-time capabilities that make it well-suited for applications demanding rapid response, such as online video comprehension.
>
> In terms of performance, VideoChat-T consistently outperforms LLaVA-OneVision on temporal grounding tasks, exhibits a slight edge on MVBench, and achieves comparable results on Egoschema. However, it demonstrates a relatively weaker performance on the VideoMME benchmark. Given the substantial computational savings afforded by VideoChat-T, we contend that this trade-off in performance on certain datasets is justifiable.
>
> **Further validation of the effectiveness and generalization of TimeSuite:**
>
> | Method                       | Charades-STA | QVHighlight | VideoMME | MLVU | MVBench |
> |------------------------------|:------------:|:-----------:|:--------:|:----:|:-------:|
> |                              |    IOU0.5    |     mAP     | w/o subs |  Avg |   Avg   |
> | Llava-OneVision (baseline) [1] |      7.3     |     15.0    |   58.2   | 64.7 |   56.7  |
> | Llava-OneVision-T (Ours)   |       42.5       |       21.7      |   61.4   | 69.4 |   56.1  |
>
> To verify the robustness of our TimeSuite for other MLLMs, we transferred our method to Llava-OneVision. The table shows the performance changes of Llava-OneVision after applying our TimeSuite. It can be seen that when we apply the full set of methods in TimeSuite to Llava-OneVision, the model's performance on two different long-video evaluation benchmarks improves (+3.2 on VideoMME and +4.7 on MLVU), effectively demonstrating the robustness of our TimeSuite for different MLLMs.
>
>
>
>
>
> **Citations:**
>
> [1] Li, Bo, et al. "Llava-onevision: Easy visual task transfer." arXiv preprint arXiv:2408.03326 (2024).

---

> ### Author Response · Authors · 2024-11-21
>
> **W2: The short-term performance on MVbench drops require a deeper investigation rather than mere explanations.**
>
>
> Thank you for your valuable suggestion. We have conducted an in-depth investigation into the performance degradation on MVBench and added the findings to Appendix F.2 (page 19) of the manuscript.
>
>
> | Method                    | post ft data           | data size | frame num | token num (per frame) | MVBench (AVG) |
> |---------------------------|------------------------|:---------:|:---------:|:---------------------:|:------------:|
> | VideoChat2                | -                      |     -     |     16     |           12          |     60.4     |
> | VideoChat2                | -                      |     -     |    128    |           12          |     42.1     |
> | VideoChat-T (Common_Init) | -                      |     -     |    128    |           3           |     25.3     |
> | VideoChat-T (Ours)        | -                      |     -     |    128    |           3           |     48.6     |
> | VideoChat-T (Ours)        | TimePro+Normal (Ours)  |   0.43M   |    128    |           3           |     59.9     |
> | VideoChat-T (Ours)        | TimePro+FullVideoChat2 |     2M    |    128    |           3           |     62.9     |
>
> **Our analysis revealed two primary factors contributing to the performance decline on MVBench:**
>
>
> - **Architectural Discrepancy:** The original VideoChat2 model was designed to process only 16 frames, leading to a mismatch in the learned feature distribution compared to the architecture of VideoChat-T. As shown in the first two rows of the table, increasing the input frame number for VideoChat2 resulted in a significant performance drop (from 60.4 to 42.1). When initializing VideoChat-T with VideoChat2, performance was close to random (25.3) due to the newly introduced randomly initialized layers. By applying efficient initialization to these new layers, we partially recovered the original capabilities of the model, bringing the MVBench performance of the un-trained VideoChat-T back to 48.6, representing an improvement of 6.5 compared to the 128-frame VideoChat2. After further fine-tuning, the short-video processing capability of VideoChat-T improved significantly, reaching 59.9.
>
> - **Fine-tuning Data Discrepancy:** We fine-tuned VideoChat-T using only 432K data, significantly less than the 2M non-grounded regular data used for training VideoChat2. The fine-tuning data for VideoChat2 primarily consisted of short videos of around ten seconds, which closely matched the length distribution of the MVBench evaluation videos, playing a crucial role in improving MVBench performance. To validate our hypothesis, we conducted additional experiments by training our VideoChat-T model using the TimePro and full VideoChat2 training data. It can be observed that VideoChat-T showed a slight improvement in performance on the MVBench dataset, achieving an accuracy of 62.9, which is an increase of 2.5 compared to the original VideoChat2.

---

> ### Author Response · Authors · 2024-11-21
>
> **W3: Instead of using implicit TAPE, consider implementing a more explicit solution like 3D RoPE (e.g., Qwen2-VL) and conduct an ablation study.**
>
> Thank you very much for your valuable suggestion. Our TAPE operates on "spatially unordered" tokens output by the QFormer, where the entire long video token sequence only differs in temporal positions without spatial variations. Therefore, applying 1D RoPE in the temporal dimension is more suitable for comparison with our method.
>
> We have supplemented our study with an ablation experiment using 1D RoPE. Specifically, we replaced our designed TAPE with 1D YaRN [2], an improved explicit 1D rotary position embedding method.
>
>
> | Method             | MVBench | VideoMME | Egoschema | Charades-STA | QVHighlight |
> |--------------------|:-------:|:--------:|:---------:|:------------:|:-----------:|
> | -                  |   Avg   |  Vision  |    Full   |    R1@0.5    |     mAP     |
> | r/w YaRN           |   59.7  |   45.6   |    59.3   |     46.9     |     25.4    |
> | VideoChat-T (Ours) |   59.9  |   46.3   |    60.0   |     48.7     |     26.5    |
>
> After implementing YaRN, the model's performance was lower than our proposed method. This is due to the higher compression rate of our visual token sequence, which significantly shortens the sequence length compared to the original length, thus limiting the effectiveness of YaRN. Our TAPE, in conjunction with Token Shuffle, is highly suitable for efficiently embedding temporal positions in the compressed long video token sequence, thereby enhancing MLLM's temporal awareness.
>
> **Citations:**
>
> [2] Peng, Bowen, et al. "Yarn: Efficient context window extension of large language models." arXiv preprint arXiv:2309.00071 (2023).

---

> ### Author Response · Authors · 2024-11-21
>
> **Q1: Could training the model on both temporal and non-temporal grounding data mitigate performance loss in short-term videos?**
>
>
>
> Your question is very valuable. To address this issue, we conducted additional ablation experiments. By training VideoChat-T with different combinations of temporal and non-temporal grounding data, we can clearly observe the impact of these two types of data on model performance. The specific results of these experiments are shown in the table below (available in Appendix G.1, page 21).
>
>
> | Method                | MVBench | VideoMME | Charades-STA |
> |-----------------------|:-------:|:--------:|:------------:|
> |                       |   Avg   | w/o subs |    R1@0.5    |
> | Normal                |   56.1  |   42.6   |      8.0     |
> | TimePro               |   57.4  |   46.0   |     45.6     |
> | TimePro+Normal (Ours) |   59.9  |   46.3   |     48.7     |
>
>
> It can be observed that the combined use of TimePro and Normal to fine-tune VideoChat-T achieves the highest performance in short video QA, long video QA, and temporal grounding tasks. This not only demonstrates that the simultaneous use of temporal and non-temporal grounding data can mitigate performance loss in short videos, but also reveals that the effects of temporal and non-temporal grounding data are complementary across various tasks. The task differences between temporal and non-temporal grounding data are very significant, effectively compensating for the model’s shortcomings from different task perspectives and feature distributions. Using both types of data can significantly enhance the overall capabilities of the model.

---

> ### Author Response · Authors · 2024-11-21
>
> **Q2: Why does temporal grounding data lead to accuracy loss in short-term videos?**
>
>
> We would like to thank you for raising thought-provoking question. We have supplemented Appendix G.2 (page 22) with an exploration of whether temporal grounding data could impair the model's original generalization capabilities.
>
> |        FT Data        | MVBench (AVG) |
> |-----------------------|:-------------:|
> |         TimeIT        |      54.7     |
> |     TimeIT+Normal     |      55.3     |
> |         Normal        |      56.1     |
> |        TimePro        |      57.4     |
> | TimePro+Normal (Ours) |      59.9     |
>
> Typically, the form of temporal grounding data is relatively singular, which can easily cause the model to overfit on temporal grounding tasks, leading to a loss of general question-answering capabilities and consequently damaging generalization. We compared the TimeIT dataset proposed in TimeChat with our TimePro on MVBench. As shown in the table, fine-tuning using only TimeIT resulted in the poorest accuracy, and the performance of the model using TimeIT+Normal was also slightly lower than that using only Normal. This indicates that TimeIT does indeed impair the model's original performance (as shown in Figure 1 at the beginning of the paper, TimeChat lost some general question-answering capabilities after fine-tuning, and the right column shows that it outputs the localization time when given a general question).
>
> In contrast, our TimePro includes diverse data, covering 15 datasets spanning 9 different task types, which can mitigate the generalization loss caused by singular data to some extent. Moreover, our data combines grounding with various general tasks. For example, Grounded Captioning requires simultaneously outputting a detailed description of the corresponding video segment, and Reasoning Temporal Localization requires reasoning over the question at the same time. This greatly enhances the model's generalization, thereby reducing the damage to the model's original capabilities (i.e., short video accuracy). As shown in the table, the performance using only our TimePro is even higher than using only Normal, and the effect of using TimePro+Normal far surpasses all other combinations. This also demonstrates that our TimePro effectively avoids impairing the model's original performance.

---

> ### Author Response · Authors · 2024-11-21
>
> **Q3: Despite this, why is short-term accuracy on VideoMME still improved?**
>
> The performance decrease of our model on MVBench and the improvement on short videos in VideoMME are due to the different definitions of short videos in the two benchmarks. The short videos in VideoMME are significantly longer than those in MVBench. The videos in MVBench are mostly around ten seconds long, often containing information from a single scene. In contrast, VideoMME defines short videos as those under two minutes, with most short videos ranging from 60 to 120 seconds and containing multiple scenes. This phenomenon further corroborates that our model performs better on longer videos.
>
>
>
>
>
> Thank you once again for your thorough review and constructive suggestions. We believe that your feedback has greatly enhanced the quality and clarity of our manuscript.

---

> > ### Comment · Reviewer_61dM · 2024-11-25
> >
> > Many thanks for the rebuttal. The additional tables provided here are so great. They solved my concerns. I will take a further check and consider to increase my rating.

---

### Public Comment · ~Shraman_Pramanick1 · 2024-11-17
**Questions about Potential Data Leakage for Charades-STA Temporal Grounding Evaluation**

Dear Authors,

Thank you for the excellent work. After reading the paper, I have a question regarding data leakage from the evaluation set during the instruction fine-tuning. I will request the authors to provide detailed clarification on this critical issue.

In Table 6, the authors mention using all 45,731 samples from the STAR [1] dataset. However, the STAR dataset is constructed with the videos from Charades' training and test set. After a quick check, I see that almost half of the videos from the Charades-STA test set appear in the STAR training set. Moreover, as the authors use **all 45,731 samples** from STAR, there is no filtering to alleviate data leakage. Hence, this leakage can be hugely responsible for the strong reported results on the Charades-STA zero-shot setting, where the authors report a massive 8.1 R1@0.7 points improvement over the previous SOTA. Though used in a different task setup, the same videos appearing in fine-tuning and evaluation would hugely affect the zero-shot performance.

It would be helpful to clarify the issue. Thanks again for your work!

[1] Bo Wu, Shoubin Yu, Zhenfang Chen, Joshua B Tenenbaum, and Chuang Gan. STAR: A benchmark for situated reasoning in real-world videos. NeurIPS, 2021.

---

> ### Author Response · Authors · 2024-11-18
>
> Thank you for your interest in our work! We have found that the STAR training set data might influence the zero-shot performance on Charades-STA. Consequently, we have added additional ablation experiments in the appendix (Table 11, Section F, page 19) to verify the impact of the STAR training set.
>
> |    Method   |        | Charades-STA |        |
> |:-----------:|:------:|:------------:|:------:|
> |             | IOU0.3 |    IOU0.5    | IOU0.7 |
> | VideoChat-T |  69.9  |     48.7     |  24.0  |
> |   w/o STAR  |  68.8  |     47.5     |  24.0  |
>
> The performance comparison indicates that removing the STAR training set does result in a slight decrease in the model's performance on Charades-STA, but it does not significantly affect the zero-shot performance. This might be because the STAR video reasoning QA task differs greatly from the temporal grounding task in Charades-STA. While video reasoning focuses on understanding the overall logic of the video, temporal grounding requires the model to accurately pinpoint the start and end times of different events within the video. This task difference means that even if there is a data leak, it doesn't provide many pre-trained temporal features for the Charades-STA task.
>
> We appreciate your valuable question, as it greatly helps us improve the quality of our manuscript. Thank you!

---

### Author Response · Authors · 2024-11-28
**Overall Comment by Authors**

We sincerely thank all the reviewers for their constructive comments. We are delighted to receive positive feedback such as "highly complete" (61dM), "well-structured" (N2GP, epk2), "easy to follow" (N2GP), "contributions are multiple folds" (61dM), "innovative approaches" (epk2), "substantial advancements" (N2GP), "significant improvements" (61dM, epk2), "systematic and effective" (x14u), "experiments are comprehensive and detailed" (N2GP, x14u), "convincing" (x14u), "broad implications" (epk2).


We have carefully addressed all the concerns raised by the reviewers point by point in the individual response section. In addition, we have supplemented our work with extensive and detailed ablation experiments to further investigate specific characteristics of our proposed method. Most of these additional experiments are included in the appendix section of the revised manuscript.


**Below is a summary of the eight new experiments added during the rebuttal phase:**

1. Comparison of the computational efficiency of VideoChat-T with other MLLMs. This shows that our model has a relatively good performance while consuming very minimal computational resources. (61dM, N2GP, M5C1, x14u, epk2)

2. Validation experiments of TimeSuite's transferability. By applying our full set of methods to LLaVA-OneVision, we observed an improvement in long video benchmarks. (61dM, M5C1)

3. In-depth exploration of the performance degradation of MVBench. This reveals two main reasons for the performance loss on MVBench. (61dM, epk2)

4. Comparison of TAPE with other position embedding methods. This verifies the superiority of TAPE for our model structure. (61dM)

5. Performance comparison of using temporal grounding and non-temporal grounding data both simultaneously and separately. This verifies the complementary functionality of temporal grounding and non-temporal grounding data. (61dM)

6. Analysis experiments on the reasons for performance loss caused by temporal grounding data. This elucidates the advantages of our data over previous temporal grounding data. (61dM, epk2)

7. Training TimeChat with our data and comparing it with our VideoChat-T. This further verifies the effectiveness of our model structure by eliminating training data interference. (N2GP, x14u)

8. Exploratory experiments on the data ratio of TimePro. This verifies the rationality of the current version of TimePro's composition. (M5C1)


We are deeply grateful to all the reviewers for their insightful comments and invaluable contributions, which have greatly improved the quality of our manuscript. If there are any remaining questions or concerns, we would be most delighted to engage in further discussions and address them to the best of our ability.

---

### Meta-Review · Area_Chair_HhbW · 2024-12-17

**Metareview:**

After discussion, this submission received 3 positive scores and a negative score. The reviewer who assigned the negative score has no response to the author’s rebuttal. After reading the paper, the review comments and the rebuttal, the AC thinks that the major concerns about technical novelty of the proposed approach and experimental evaluation were comprehensively solved. The AC supports the acceptance of this submission as a poster paper in ICLR.

**Additional Comments On Reviewer Discussion:**

After discussion, three of reviewers thought that their concerns were solved and raised the score. For example, authors added short-term performance on MVbench, which is raised by reviewer 61dM. For the technical novelty issue raised by reviewer N2GP and x14u, the authors listed `structure novelty`, `data novelty` and `valuable findings` as responses. The reviewer who assigned the negative score has no response to the author’s rebuttal.

---

### Decision · Program_Chairs · 2025-01-22

Accept (Poster)